# β2 and β3a regulatory subunits can coassemble in the same BK channels

Yu Zhou[1], Vivian Gonzalez-Perez[1], Xiao-Ming Xia[1], Gopal S. Kallure[2], Sandipan Chowdhury[2], and Christopher J. Lingle[1]

Ca²⁺- and voltage-activated BK-type K⁺ channels are influenced profoundly by associated regulatory subunits, including β subunits (*Kcnmb1–4*; β1–β4). Although overlap in expression of different BK β subunits occurs in native tissues, whether they can coassemble in the same channel complex is not known. We coexpress β2 and β3a subunits together with BK α and, through a combination of macroscopic and single-channel recordings, along with quantitative pull-down of tagged subunits, test whether coassembly can occur. We evaluate two models: (1) random mixing in which β2 and β3a subunits coassemble in the same channels, and (2) segregation in which β2 and β3a are found in separate complexes. Our results support the view that, for β2 and β3a, BK currents arise from the random, independent assembly of both subunits in the same channels. Single-channel recordings directly confirm coassembly of β2 and β3a subunits in the same channels. Quantitative biochemical analysis of coexpression of tagged β2, β3a, and BK α subunits also reveals that β2:β3a:α ternary complexes form.

## Introduction

Ca²⁺- and voltage-dependent large conductance Ca2⁺-activated K⁺ channel (BK)-type, large-conductance K⁺ channels exhibit extensive functional diversity that arises in part from tissue-specific expression of regulatory subunits (Gonzalez-Perez and Lingle, 2019) that coassemble with the pore-forming α subunit encoded by a single KCNMA1 gene (Butler et al., 1993; Salkoff et al., 2006). Three distinct families of regulatory subunits have been identified to date (β, γ, and LINGO) (Li and Yan, 2016; Gonzalez-Perez and Lingle, 2019; Dudem et al., 2020). Of these three families, the physiological roles of various β subunit isoforms have probably received the most attention, but many important questions remain unaddressed. The β subunit family (β1–β4) (Knaus et al., 1994a; Wallner et al., 1999; Xia et al., 1999; Brenner et al., 2000; Meera et al., 2000; Uebele et al., 2000; Weiger et al., 2000; Xia et al., 2000) includes four homologous mammalian genes (human: KCNMB1–KCNMB4; mouse: Kcnmb1–Kcnmb4), and the expressed subunits each uniquely influence functional properties of the resulting BK channels. This includes the range of voltages over which BK channels activate at a given Ca²⁺ (Nimigean and Magleby, 1999; Cox and Aldrich, 2000), the kinetics of activation and deactivation (Cox and Aldrich, 2000; Orio and Latorre, 2005), and the kinetics of BK channel inactivation onset and recovery (Wallner et al., 1999; Xia et al., 1999; Xia et al., 2000; Zeng et al., 2007). β subunits also impact such properties as single-channel current rectification (Zeng et al., 2003) and pharmacology (Xia et al., 1999; Meera et al., 2000; Zeng et al., 2003). Furthermore, the functional diversity arising

from β subunits is influenced by the stoichiometry of assembly of a given β subunit (i.e., 0–4 β subunits per channel) in a BK complex (Ding et al., 1998; Wang et al., 2002); one unexplored question is whether two different β subunits can assemble in the same BK complex, in cells in which multiple types of β subunits may be expressed.

Despite the availability of screening techniques for RNA in different tissues and cells, there is little information regarding the extent to which coexpression of different β subunits may occur in the same cells. Although the Allen Brain Atlas qualitatively suggests regions of β subunit overlap (https://mouse.brain-map.org/experiment/show?id=81600586), one challenge is that message levels for some β subunits, as assessed by quantitative PCR in some tissues/cells, are remarkably low even in cases where BK currents may clearly exhibit the functional signature conferred by a given β subunit, e.g., β2 in mouse adrenal chromaffin cells (Martinez-Espinosa et al., 2014). To our knowledge, only one paper has proposed, based on functional criteria, the possible presence of two distinct β subunit isoforms. Specifically, based on functional and pharmacological criteria, cerebellar Purkinje cells have been proposed to express both β2-containing and β4-containing BK channels, perhaps segregated into different parts of the cell (Benton et al., 2013). Although it might be assumed that different BK β subunits should be able to coassemble in the same channels, this has not been tested, and the cerebellar results raise the possibility that segregation is possible. Recently resolved cryo-EM structures of the BKα/β4

[1]Department of Anesthesiology, Washington University School of Medicine, St. Louis, MO, USA; [2]Department of Molecular Physiology and Biophysics, The University of Iowa, Carver College of Medicine, Iowa City, IA, USA.

Correspondence to Christopher J. Lingle: clingle@wustl.edu; Yu Zhou: zhouy@wustl.edu.

complex (Tao and MacKinnon, 2019) have unveiled extensive inter-subunit contacts between the extracellular loops of adjacent β4 subunits, implying a potential symmetric requirement when multiple β subunits coassemble with BKα. As the extracellular loop structure may vary among different BK β subunits (Zeng et al., 2003), it is important to ascertain whether diverse β subunits can coassemble in a single BK channel and affect the function of the resulting channel. Here, we have tested the consequences of heterologous coexpression of β2 and β3a BK β subunits.

Taking advantage of distinct functional differences conferred by β2 and β3a subunits on BK channels, we use a combination of macroscopic current recordings, single-channel recordings, and quantitative biochemical evaluation of complexes to ask whether β2 and β3a subunits coassemble in single BK channel complexes. Both β2 and β3a subunits produce kinetically similar rates and extent of inactivation. However, they differ markedly in properties of tail currents during recovery from inactivation. For β2-mediated inactivation, there is almost no tail current over a time period that results in full recovery from inactivation (Solaro et al., 1997; Xia et al., 1999). In contrast, for β3a-mediated inactivation, a prolonged slow tail current is observed that results in a net current flux that exceeds by several fold that expected for a channel population recovering in accordance with a simple open channel block behavior (Zeng et al., 2007; Gonzalez-Perez et al., 2012). By determining the maximal BK current in a patch (which can be done by rapid removal of inactivation by trypsin), the slow tail currents provide a measure of the fraction of channels that have inactivated via a β3a N terminus. Using measurements of changes in tail current amplitude and the fraction of channels that are non-inactivating at different time points during trypsin digestion, we develop models for expectations based on either segregated assembly of β2 and β3a subunits vs. random mixing of subunits. The analysis depends on the application of a trinomial distribution model to evaluate predicted behaviors arising from the random mixing model, in comparison to the sum of two binomial distributions to describe the segregated assembly model. The results provide strong support for the idea that the distribution of β2 and β3a subunits among BK channels, when coexpressed in *Xenopus* oocytes, results from random assembly into the four β subunit–binding positions in each BK channel. Single-channel results, although not being sufficient to test for independence of subunit assembly, provide direct support that both β2 and β3a subunits can be present in the same BK channels, likely reflecting the full set of possible stoichiometries. Finally, quantitative biochemical separation following coexpression of tagged BK α, hβ2, and hβ3a subunits directly supports the idea that ternary (α:β2:β3a) complexes, of different stoichiometries, do form. Together, these results argue that different β subunits can coassemble in the same BK complexes, adding additional nuance to the potential physiological diversity that BK channels can adopt in native cells.

## Materials and methods
### Channel expression and molecular biology
Stage IV *Xenopus* oocytes were obtained from *Xenopus* 1. BK channels were expressed by cRNA injection of the pore-forming mouse Slo1 (mSlo1) α-subunit alone or with human β2 (hβ2), mouse β3a (mβ3a), or mixtures of hβ2:mβ3a in defined ratios. Mouse β3a was preferred over human β3a because mβ3a produces faster and more complete inactivation and more prolonged tail currents compared with hβ3a, each feature being advantageous for the experiments described here (Zeng et al., 2008). cRNAs were injected at nominal ratios by weight. In preliminary tests with hβ2:mβ3a coexpression, we found that equimolar β2:β3a injection ratios yielded currents similar to those with β2 alone. Thus, β2:β3 injection ratios needed to be strongly biased toward higher mβ3a quantities to obtain clear indications of the presence of channels containing mβ3a. Yet, lower quantities of mβ3a cRNA coinjected with α alone successfully resulted in robust expression of β3a-containing channels when expressed in the absence of β2. Because of this, we tested changes in the amount of BK α expressed with different β2:β3a ratios and also remade cRNA multiple times without identifying any conditions that increased relative effectiveness of mβ3a assembly into channels when coexpressed with hβ2. Therefore, for experiments in which we have quantitatively evaluated the impact of different hβ2:mβ3a nominal expression ratios, both hβ2 and mβ3a cRNA were prepared in parallel with the same reagents, and tests of different ratios were all done from the same stock solutions and in the same batch of oocytes. For coexpression of Slo1 with hβ2, we used a fixed molar ratio of hβ2:mSlo1α of 4:1. For coexpression of Slo1: hβ2:mβ3a, the hβ2: mSlo1 ratio was fixed at 1:1, with mβ3a added to produce the hβ2:mβ3a ratios specified in the text. Under these conditions, this insured that there would always be a quantity of hβ2 sufficient to produce near full stoichiometric occupancy of β subunit sites of interaction.

### Electrophysiology
During seal formation, oocytes were bathed in a standard frog ringer (in mM): 115 NaCl, 2.5 KCl, 1.8 CaCl$_2$, and 10 HEPES, at pH 7.4. Inside-out patches from *Xenopus* oocytes were used in all experiments. Currents were recorded using an Axopatch 200B amplifier (Molecular Devices), low pass-filtered at 10 kHz, and digitized at 100 kHz. Following excision, patches were quickly moved into a flowing low Ca$^{2+}$ solution. The pipette/extracellular solution contained (in mM): 140 K-methanesulfonate, 20 KOH, 10 HEPES, and 2 MgCl$_2$, adjusted to pH 7 with methanesulfonic acid. The cytosolic face of the excised patches was perfused with intracellular solutions containing either 0 or 10 μM Ca$^{2+}$ with the following (in mM): 140 K-methanesulfonate, 20 KOH, and 10 HEPES, pH adjusted to 7.0 with methanesulfonic acid, and one of the following: 5 EGTA (for nominally 0 Ca$^{2+}$) or 5 HEDTA with added Ca$^{2+}$ to make 10 μM Ca$^{2+}$. A commercial set of Ca$^{2+}$ standards (WPI) was used for calibration of Ca$^{2+}$ concentrations with a Ca$^{2+}$-sensitive electrode (Orion, Thermo-Fisher Scientific).

Experiments in which trypsin was used to remove BK inactivation were as previously reported (Zhang et al., 2006; Zhang et al., 2009). To summarize, a 0 Ca$^{2+}$ solution containing 0.05 mg/ml trypsin (cat #T0303-1G; Sigma-Aldrich) was applied for timed intervals, ranging from 2.5 to 20 s. Both before and after each timed trypsin application, the patch was washed with enzyme-free 0 Ca$^{2+}$ solution for 5 s. Between 0 Ca$^{2+}$ applications, the patch was exposed to 10 μM Ca$^{2+}$ solution, during which

protocols to test for BK current properties were applied. Plots of trypsin removal of inactivation reflect the cumulative time a patch was exposed to a given trypsin concentration. An SF-77B fast perfusion stepper system (Warner Instruments) was used to switch among solutions and was controlled via pClamp software (Molecular Devices). Experiments were at room temperature (~22–25°C). All chemicals were obtained from Sigma-Aldrich.

G-V relationships were determined from tail currents measured 150 μs after repolarization. The Boltzmann function to fit the G-V curves was

$$G(V) = \frac{G_{max}}{1 + e^{-\frac{z(V-V_h)}{kT}}},\qquad(1)$$

in which $G_{max}$ is maximal conductance, $z$ is apparent voltage dependence in units of elementary charge, and $V_h$ is the voltage of half-maximal activation, with $kT$ set to 25.7 meV for 25°C. Slow tail amplitude arising from β3a tails was assessed by the amplitude at 500 μs after the nominal repolarizing step to minimize any contribution of the fast tail of BK channels that close after inactivation has been removed. For single-channel patches, leak and capacitance currents were subtracted by using blank traces from the same recording.

Data were analyzed using OriginPro 7.5 (OriginLab Corporation), Clampfit (version #9.2.1.9) (Molecular Devices), Microsoft EXCEL, or programs developed in this laboratory. Error bars in the figures represent SDs. Each measurement is averaged from at least four experiments.

## Models of subunit assembly

Specific details explaining the application of binomial and trinomial distributions for analysis of β2 and β3a subunit composition in populations of BK channels are developed in the Results. Here, the basic underpinnings of our approach are provided. The application of a binomial distribution applies to cases where a trial (or observation) can have two possible outcomes with the distribution function given by

$$P(x) = \frac{n!}{x!(n-x)!}p^x(1-p)^{n-x}.\qquad(2a)$$

For application to a tetrameric channel population, with $n = 4$ potential sites of occupancy but individual sites may either be occupied by a β subunit or be vacant, P(x) returns the probability that, in the population, channels will have x = 0–4 β subunits, with $p$ being the probability of occupancy in the channel fraction. Probability of occupancy in the channel population can be conveniently termed the mole fraction of sites occupied by a given β subunit. For the often-used example in which P = 0.5, i.e., half the available β subunit occupancy positions are occupied, with channels containing 0–4 subunits given by P(0) = 0.0625, P(1) = 0.25, P(2) = 0.375, P(3) = 0.25, and P(4) = 0.0625. Guided by early work on the stoichiometry of Shaker K channel assembly (MacKinnon, 1991) and inactivation (MacKinnon et al., 1993), for BK channels, the binomial distribution has been previously applied to several distinct cases: (1) evaluation of cases in which the mole fraction of β subunits expressed in a cell is less than the available sites of occupancy on the pore-forming subunit population (Ding et al., 1998); (2) for description of changes in

stoichiometry during partial removal of inactivation domains (IDs) from a channel population via trypsin, thereby allowing estimations of the fraction of channels that may still have inactivating N termini (Ding et al., 1998); and (3) for evaluation of assembly of WT and the G375R channelopathy mutant (Geng et al., 2023). For present purposes, if β2 and β3a subunits randomly mix among channels, the predicted distributions for β2:β3 occupancies are determined exactly as given in Eq. 2a, but as

$$P(x, n-x) = \frac{n!}{x!(n-x)!}p_{\beta2}^x(1-p_{\beta2})^{n-x},\qquad(2b)$$

with the mole fraction of β3a, $p_{\beta3a} = 1-p_{\beta2}$. For the simple case of $p_{\beta2} = p_{\beta3a} = 0.5$, P(0β2:4β3a) = 0.0625, P(1β2:3β3a) = 0.25, P(2β2:2β3a) = 0.375, P(3β2:1β3a) = 0.25, and P(4β2:0β3a) = 0.0625.

If β2 and β3a subunits are randomly mixed among channels, with each independently competing for occupancy of a common set of four positions of association, when all sites on the BK α subunit are occupied, that initial distribution will also be binomially distributed, reflecting relative probabilities of occupancy of the two subunits with $p_{\beta2}+ p_{\beta3a} = 1$. However, once one is trying to assess a situation in which inactivation can be removed, there are now three possibilities for each position of occupancy: intact β2, intact β3a, and subunits with N terminus removed (IR), Since inactivation or its absence is the reporter of functional contributions of each subunit, we assume that a non-inactivating subunit, whether β2 or β3a, does not matter for this analysis. Such a situation is described by a trinomial distribution, which is given by

$$P(X = x, Y = y) = \frac{n!}{x!y!(n-x-y)!}p_{\beta2}^x p_{\beta3a}^y(1-p_{\beta2}-p_{\beta3a})^{n-x-y},\qquad(3)$$

with x = 0,1..n, y = 0,1...n, x+y<n, and where $p_{\beta2}$ would correspond to the probability of occupancy by β2, $p_{\beta3a}$ to the probability of occupancy by β3a, and 1- $p_{\beta2}$ - $p_{\beta3a}$ to the probability that a subunit is now IR, with P(X,Y), giving the overall fractional likelihood for all 15 stoichiometric combinations (see Fig. 4).

## Evaluation of the time course of removal of inactivation by trypsin

In previous work, the time course of removal of inactivation by trypsin was measured for both WT and mutant hβ2 (Zhang et al., 2006; Zhang et al., 2009) and also WT and mutant mβ3a (Zhang et al., 2009), permitting inferences regarding the accessibility of trypsin-sensitive residues on the N-terminal segments. The time course of digestion could be reliably described for a given construct by the following function:

$$I_{ss}(t)/I_{max} = \left(1 - e^{-\frac{t}{\tau}}\right)^n,\qquad(4)$$

with $I_{ss}$ reflecting the amplitude of non-inactivating current at different time points during trypsin digestion, $I_{max}$ reflecting the steady-state current after digestion is complete, τ being the time constant of digestion, and $n$ reflecting the idea that all N termini must be removed by trypsin in order for inactivation to be fully abolished.

Here, we outline expectations for removal of inactivation by trypsin for segregated and random mixing models. For a BK

population in which β2 and β3a subunits are segregated, the simple expectation, derived from Eq. 4, is

$$I_{ss}(t)/I_{max} = f_{\beta2}\left(1 - e^{-\frac{t}{\tau_{\beta2}}}\right)^4 + f_{\beta3a}\left(1 - e^{-\frac{t}{\tau_{\beta3a}}}\right)^4, \quad (5)$$

where $f_{\beta2}$ and $f_{\beta3a}$ are the respective starting fractions of each isoform within the total channel population. We make the initial assumption that the total amount of injected β2 and β3a cRNA under all injection conditions is sufficient to fully occupy all available β subunit sites of occupancy on the BK α subunits. To facilitate understanding of our approach below that utilizes a trinomial distribution to consider random mixing of β2 and β3a subunits, here, we show that Eq. 5, at t = 0, can be trivially recast in accordance with the behavior predicted for the sum of two binomial distributions. At t = 0, the $f_{\beta2}$ population begins with 4 β2 subunits per channel ($p_{\beta2}(t = 0) = 1$), while the $f_{\beta3a}$ fraction has 4 β3a subunits per channel ($p_{\beta3a}(t = 0) = 1$). For each point in time, the mole fractions of either β2 or β3a are reduced in accordance with the measured time constants of digestion of single N termini defined, respectively, by

$$p_{\beta2}(t) = p_{\beta2}(0)e^{-t/\tau_{\beta2}} \quad (6a)$$
$$p_{\beta3a}(t) = p_{\beta3a}(0)e^{-t/\tau_{\beta3a}}. \quad (6b)$$

Thus, based on the binomial distribution, for the β2-containing population,

$$P_{\beta2}(x, t) = \frac{n!}{x!(n-x)!}p_{\beta2}(t)^x\left(1 - p_{\beta2}(t)\right)^{n-x} \quad (7a)$$

and, similarly for the β3a-containing population,

$$P_{\beta3a}(y, t) = \frac{n!}{y!(n-y)!}p_{\beta3a}(t)^y(1 - p_{\beta3a}(t))^{n-y}. \quad (7b)$$

For the overall population of channels, the predicted distribution of occupancies is given by the sum of Eq. 7a and Eq. 7b, yielding

$$P(x, n-x) + P(y, n-y) = \frac{n!}{x!(n-x)!}p_{\beta2}^x(1 - p_{\beta2})^{n-x}$$
$$+ \frac{n!}{y!(n-y)!}p_{\beta3a}^y(1 - p_{\beta3a})^{n-y}. \quad (8)$$

Eq. 7a and Eq. 7b give, respectively, the changes in distributions of intact β2 subunits and IR-β2 subunits, and the same for the β3a-containing population, as trypsin digestion proceeds. Eq. 8 essentially defines the changes, based on two bionomial populations, in the expected fraction of channels of any starting composition to become non-inactivating (steady-state current). Similarly, changes in the fraction of channels that will have β3a-mediated slow tail current can be determined.

Since a channel only becomes non-inactivating when all four IDs have been removed, this corresponds to x = 0 in Eq. 7 a and y = 0 in Eq. 7 b. As such, the time course of appearance of IR-only channels for the β2-containing population, where x = 0 and n = 4, is explicitly given by

$$P_{\beta2}(0, t) = \left(1 - p_{\beta2}(0)e^{-t/\tau_{\beta2}}\right)^4. \quad (9)$$

Similarly, for β3a-containing channels,

$$P_{\beta3a}(0, t) = \left(1 - p_{\beta3a}(0)e^{-t/\tau_{\beta3a}}\right)^4. \quad (10)$$

For removal of inactivation for two segregated fractions ($p_{\beta2}$ and $p_{\beta3a}$ with $p_{\beta2}+p_{\beta3a} = 1$) of β2 and β3a subunits in the population, the overall digestion time course is given by

$$I(t) = p_{\beta2}(0)\left(1 - e^{-t/\tau_{\beta2}}\right)^4 + (1 - p_{\beta2}(0))\left(1 - e^{-t/\tau_{\beta3a}}\right)^4. \quad (11)$$

These equations are identical to the direct analytical descriptions given in Eqs. 4 and 5 to describe removal of inactivation and presumably can be used to fit predictions of digestion time course at different mole fractions of β2 and β3a subunits.

These considerations form the underpinnings of our evaluation of changes when considering a model with random mixing of β2 and β3a subunits in a population of channels. Assuming that all sites to which β subunits can associate with BK α subunits are occupied following coinjection of both β2 and β3a subunits, if both subunits assemble randomly and independently, the initial distribution of different assemblies (4:0, 3:1, 2:2, 1:3, 0:4) is defined by a binomial distribution. However, as soon as N termini are removed, whether for β2 or β3a, that defines a third category, those with a functional N terminus, given by $1-p_{\beta2}-p_{\beta3a}$.

$$P(X = x, Y = y) = \frac{n!}{x!y!(n-x-y)!}p_{\beta2}^x p_{\beta3a}^y(1 - p_{\beta2}-p_{\beta3a})^{n-x-y}. \quad (12)$$

With the assumption that both β2 and β3a IDs are randomly and independently removed by trypsin and that channels are only fully inactivated-removed (IR) when both x = 0 and y = 0, the time course of appearance of channels in P(0,0), i.e., the removal of inactivation, is defined by

$$P(0, 0, t) = p_{\beta2}^0 p_{\beta3a}^0(1 - p_{\beta2}(t)-p_{\beta3a}(t))^4, \quad (13)$$

leading to

$$I(t)/I_{max} = (1 - p_{\beta2}(0)e^{-t/\tau_{\beta2}} - p_{\beta3a}(0)e^{-t/\tau_{\beta3a}})^4. \quad (14)$$

## Impact of differential inactivation likelihood on calculations of fractional slow tail

Changes in slow tail current behavior arising from inactivation mediated by the β3a N terminus depend not only on the number of β3a N termini in a given channel but also, for channels in which both β2 and β3a N termini are present, on the relative likelihoods ($L_{\beta2}$, $L_{\beta3a}$) that each type of N terminus may contribute to inactivation at the end of a depolarization. For any given channel with x β2 N termini and y β3a N termini (P(x,y), when $L_{\beta3a} = L_{\beta2}$, the likelihood that a channel with a given β2:β3a stoichiometry will be inactivated by β3a ($pi_{\beta3a}$) at the end of a depolarization is then given by $pi_{\beta3a} = yL_{\beta3a}/(yL_{\beta3a}+xL_{\beta2})$. For cases in which the relative likelihood is not identical and $L_{\beta3a} = kL_{\beta2}$, then $pi_{\beta3a} = yL_{\beta3a}/(yL_{\beta3a}+xL_{\beta2}/k)$. For channels of P(3,1), P(2,2), and P(1,3) with $L_{\beta3a} = L_{\beta2}$, $pi_{\beta3a} = 0.25$, 0.5, and 0.75, respectively, while for $L_{\beta3a} = 3L_{\beta2}$, $pi_{\beta3a} = 0.5$, 0.75, and 0.9.

In assessing the behavior of single channels arising from hβ2:mβ3a coexpression, similar considerations were used to make estimates of likely hβ2:mβ3a stoichiometry that might underlie

the observed fractional occurrence of mβ3a-type tail current behavior (Fig. 8 C).

## Biochemical tests of expression and assembly
For protein expression, we used a modified human Slo1 construct, featuring a truncation in the RCK1–RCK2 linker and the C-terminal end, to improve biochemical behavior, used previously for single-particle cryo-EM reconstructions of hSlo1 (Tao and MacKinnon, 2019). This construct (hSlo1$_{EM}$) was tagged on the C terminus with a twin-strep tag with an intervening 3C-protease cleavage site. Full-length human β2 and human β3a genes were C-terminally tagged with mCherry-FLAG and eGFP-ρ1D4 tags, separated from the C terminus via a 3C-protease cleavage site. Plasmid DNA of these expression constructs, in pEG vectors, was isolated on a large scale using endotoxin-free Megaprep kits (Qiagen) and used for protein expression.

For the 1:1:1 (TR1) transfections, 300 ml suspension cultures of HEK293F cells, maintained in Gibco Freestyle 293 media, supplemented with 2 % Heat-Inactivated FBS (Gibco), were mixed with 225 µg each of Slo1-TS, β2-RF, and β3-Gρ, pre-incubated for 10 min at room temperature, with 2.025 mg of PEI (linear 25 kDa) in 30 ml of OptiMEM media. For 1:0.2:1.8 (TR2) transfections, the total weights of plasmids used for transfection were 225 µg of Slo1-TS, 45 µg of β2-RF, and 405 µg of β3-Gρ. After transfection, cells were grown at 37°C for 10–14 h, after which sodium butyrate was added to a final concentration of 10 mM, transferred to 30°C, and grown for another ~55 h. Cells were pelleted, washed with PBS, and frozen until use. Each 300 ml culture yielded 4–5 g of wet pellets.

Frozen cell pellets were thawed on ice, resuspended in 20 ml of chilled buffer (1 M KCl, 100 mM Tris-Cl, 40 % vol/vol glycerol, 10 mM CaCl$_2$, and 10 mM MgCl$_2$, pH 7.8), and sonicated to generate a homogenous suspension. 20 ml of 2 % w/v digitonin was added to the homogenate, and the mixture was gently agitated at 4°C for 1.5 h. The lysate was spun at 100,000 g for 45 min 20–40 µl of the supernatant was preserved for fluorescence size exclusion chromatography (FSEC) analysis, and the remainder (~40 ml) was incubated with 0.5-ml streptactin XT resin for 12–16 h. The resin was pelleted by spinning at 3,000 rpm for 10 min and subsequently washed five times, each time with 4 ml of wash buffer (WB: 500 mM KCl, 50 mM Tris-Cl, 20 % vol/vol glycerol, 0.1 % wt/vol digitonin, 10 mM CaCl$_2$, and 10 mM MgCl$_2$, pH 7.8). Total protein was eluted in 4 ml of WB supplemented with 50 mM biotin. 2 ml of the total Slo1 isolated was incubated with 50 µl of anti-FLAG M2 antibody resin (Sigma-Aldrich) or 50 µl anti-ρ1D4 resin (Cube Biotech) for 6 h at 4°C with gentle shaking. The resin was washed three times with 5× resin volume of WB and eluted in a total of 450 µl of WB supplemented with 0.2 mg/ml 3× FLAG peptide (anti-FLAG resin) or 450 µl of WB supplemented with 1 mM ρ1D4 peptide (3 steps, incubating the resin each time with 150 µl of elution buffer for ~1 h). 400 µl of the FLAG-eluted protein was incubated with 50 µl equilibrated anti-ρ1D4 resin. Before incubation, the resin was thoroughly depleted of equilibration buffer (WB) by centrifuging the washed resin in spin columns (14,000 rpm for 1 min). After 4 h of incubation, the flow-through was collected by spin filtration, and the volume of the recovered flow-through was verified to ensure that it was within 95% of the input volume. Resin-bound protein was washed and eluted as described. Similarly, the ρ1D4-eluted protein (from the second step affinity stage) was purified via anti-FLAG affinity purification.

For FSEC analysis, 20–60 µl of protein fraction was injected into a Superose 6 Increase, 10/300 Column (GE) using a buffer of composition: 300 mM KCl, 20 mM Tris, pH 7.8, and 0.1 mM GDN (Anatrace) operating at a flow-rate of 0.4 ml/min. The chromatography profiles were monitored using RF-20Axs detector (Shimadzu) using Ex./Em. Wavelengths of 587/610 nm (mCherry) and 488/507 nm (eGFP). Except for the total cellular lysates, all fractions tested via FSEC were largely monodisperse with a dominant peak at ~12.8–13.2 ml, with a full-width at half maxima of ~1 ml, which corresponded to Slo1 complexes. Such peaks, in the mCherry and eGFP channels, were fitted to a single Gaussian curve, and the area under the fitted curve was used for the ρGC calculations and evaluations of the species fractions. Uncomplexed β subunits, substantially smaller in size, elute at a retention volume of ~16 ml. For the cellular lysates, the chromatograms were polydisperse, particularly in the eGFP channel (corresponding to β3-Gρ), and we numerically integrated the area under the discretized chromatogram to assess the total species amount

$$Area = \sum_{V=8.5\,ml}^{17.6\,ml} (V_{i+1} - V_i)(C_{i+1} + C_i),$$

where $V_i$ and $C_i$ are the $i^{th}$ point on the chromatogram ($V_i$ being the volume and $C_i$ being the magnitude of the signal), and the integration is performed spanning the profile, after void peak and before the free/cleaved mCherry/eGFP peaks. For our instrumentation, the ratio of the eGFP:mCherry intensities for equimolar amounts of proteins were experimentally determined to be 19.6 and were used to convert eGFP:mCherry fluorescence intensity ratios (ρGC) into molar ratios. The experiments were performed in triplicates (three-independent transfections of each TR).

## Statistics
All results that report the mean and SD of some parameter are derived from some number (n) of patches collected from separate oocytes. When averaged data were fit with some function, parameter values are reported along with 90% confidence limits of the fit. However, statistical comparisons done on parameter values involve comparisons of individual parameter estimates, each from a distinct patch. All experiments that evaluate different ratios of injected β2 and β3a subunits utilized a single batch of β2 RNA and single batch of β3a RNA. When the ability of different models to fit data is compared, e.g., the time course of removal of inactivation for different β2:β3a injection ratios, we determined sum-of-squares not only for the entire data set but also for each β2:β3a ratio, allowing a better evaluation of whether a given model improves the quality of fit over all experimental conditions.

## Online supplemental material
Fig. S1 demonstrates that slow β3a-mediated tail current time constants do not change as the average fraction of β3a subunits

in a BK complex is reduced. Fig. S2 shows that differential likelihoods that a β3a vs. β2 N terminus will produce inactivation are unlikely to confound tests between the random mixing vs. segregated models. Fig. S3 show that partial occupancies of BK channels by β3a and/or β2 subunits are unlikely to confound the ability to distinguish between random mixing vs. segregated models. Fig. S4 compares global fits of the random mixing model vs. segregated model of the time course of trypsin-mediated removal of inactivation over 6 subunit injection conditions, utilized different constraints on model parameters. In all cases, the random mixing model yields better fits.

## Results

### Differential inactivation properties of β2 and β3a subunits as reporters for coassembly

Biochemical (α+β1) (Knaus et al., 1994b) and structural (α+β4) (Tao and MacKinnon, 2019) work has indicated that BK pore-forming α subunits assemble in a complex of up to 4:4 with BK β subunits. A similar conclusion was reached based on functional properties of heterologously expressed inactivating α:β2 (and α:β1–β2NT) channels (Wang et al., 2002), with the nuance that functional BK complexes with less than a full complement of β subunits could also occur, a phenomenon that can also occur in native cells (Ding et al., 1998). Here, we address whether, when two β subunit isoforms are expressed in the same cell, they segregate (Segregated Model) into different BK channel complexes or can they coassemble (random mixing model) in the same complexes? We utilize three distinct approaches to address this question: first, the use of macroscopic current properties in patches; second, single-channel recordings; and, third, biochemical tests utilizing heterologous expression of tagged subunits. We begin with the macroscopic current approach, since this potentially permits a quantitative evaluation of the stoichiometry of assembly of different variants of β subunits within a large population of channels.

The macroscopic approach is predicated on the idea that, for two β subunit isoforms that inactivate with different properties, perhaps differences in inactivation behavior might serve as a reporter for the presence or absence of the fraction of channels that become inactivated by a given β subunit isoform in the channel population. Here, we summarize properties of inactivation mediated by the β2 and β3a isoforms when expressed alone that are critical for this approach. In each case, we presume that, of the four IDs per channel, inactivation arises from binding of one ID per channel, and only one ID can bind at a time (Wang et al., 2002). Thus, the rate of inactivation onset resulting from a channel with four intact IDs is four times that of a channel with only a single intact ID. Another important feature for the present analysis is that recovery from BK β-mediated inactivation is defined by dissociation of the single ID that has produced inactivation (Ding et al., 1998), irrespective of the number of β subunits in the channel. Both of these properties of BK β-mediated inactivation are shared with inactivation of the ShB Kv channel (MacKinnon et al., 1993). These characteristics of BK inactivation have been clearly established for β2-mediated inactivation but also appear to hold for β3a-mediated inactivation.

Macroscopically, the onset of mouse β3a-mediated inactivation is about two to three fold faster than β2-mediated inactivation (Fig. 1, A, B, and E). Despite the general similarity in the onset of β2 and β3a inactivation, they differ markedly in terms of tail currents following repolarizations that lead to recovery from inactivation. β3a-mediated inactivation is associated with a prolonged slow tail of current (Fig. 1 A) that opens essentially instantaneously upon repolarization (Zeng et al., 2007; Gonzalez-Perez et al., 2012; Zhou et al., 2025). In single-channel traces, repolarization of α+β3a channels results in immediate opening to an opening level of reduced conductance compared with normal BK opening (Fig. 1 C, bottom). This behavior has been explained by the idea that β3a inactivation involves entry first into a preinactivated open state (O*), which is in rapid voltage-dependent equilibrium with an inactivated state (I) (Zeng et al., 2007; Gonzalez-Perez et al., 2012; Zhou et al., 2025). This delays the return to a fully O state and delays recovery from inactivation. In contrast, no detectable tail current is observed during recovery from β2-mediated inactivation, either in macroscopic currents (Fig. 1 B) or in single-channel traces (Fig. 1 D). The differences in tail currents provide one of the key tools used here to evaluate the fractions of β2 and β3a N termini present in a channel population when the two variants are coexpressed.

The second critical feature necessary for our tests is that both β2- and β3a-mediated inactivation can be removed quantitatively by carefully timed applications of cytosolic trypsin (Fig. 2, A and B). During digestion by trypsin, the inactivation time constant slows approximately fourfold as N termini are removed, while the fraction of non-inactivating current increases. As described in the Materials and methods, channels only become fully non-inactivating when all N termini are digested. As a consequence, the digestion time course for both α+β2 and α+β3a current is described by $I(t)=(1-exp(-t/\tau))^4$ (Fig. 2 C). The ability of this function to describe the digestion time course in both cases supports the idea that each channel complex arises from the presence of close to 4 β subunits, each with an intact ID prior to trypsin application, and that all four N termini must be digested to remove inactivation. That a function raised to the fourth power fits adequately also supports the idea that most channels in these patches are fully occupied by 4 β subunits, whether β2 or β3a. Consistent with earlier results (Zhang et al., 2006; Zhang et al., 2009), the rate of removal of inactivation differs markedly between α+β3a and α +β2, with digestion about sevenfold slower for β2. This potentially provides a tool to differentially alter the relative fractions of β2 and β3a N termini in the channel population.

If the β3a-mediated slow tail directly reports on the fraction of channels that have inactivated via a β3a N terminus, trivially the amplitude of the slow tail during digestion by trypsin should be inversely related to the fraction of channels that become non-inactivating ($f_{ss}$) at different time points of digestion as observed in Fig. 2 D. In contrast, during digestion of α +β2 current, at the time point at which the β3a slow tail amplitude is readily measured, e.g., 3 ms, there is no change in tail current during α+β2 digestion (Fig. 2 D). For each patch, digestion by trypsin permits normalization of the slow tail amplitude to the total BK conductance in the patch. It should be noted (Fig. 2 B) that the initial

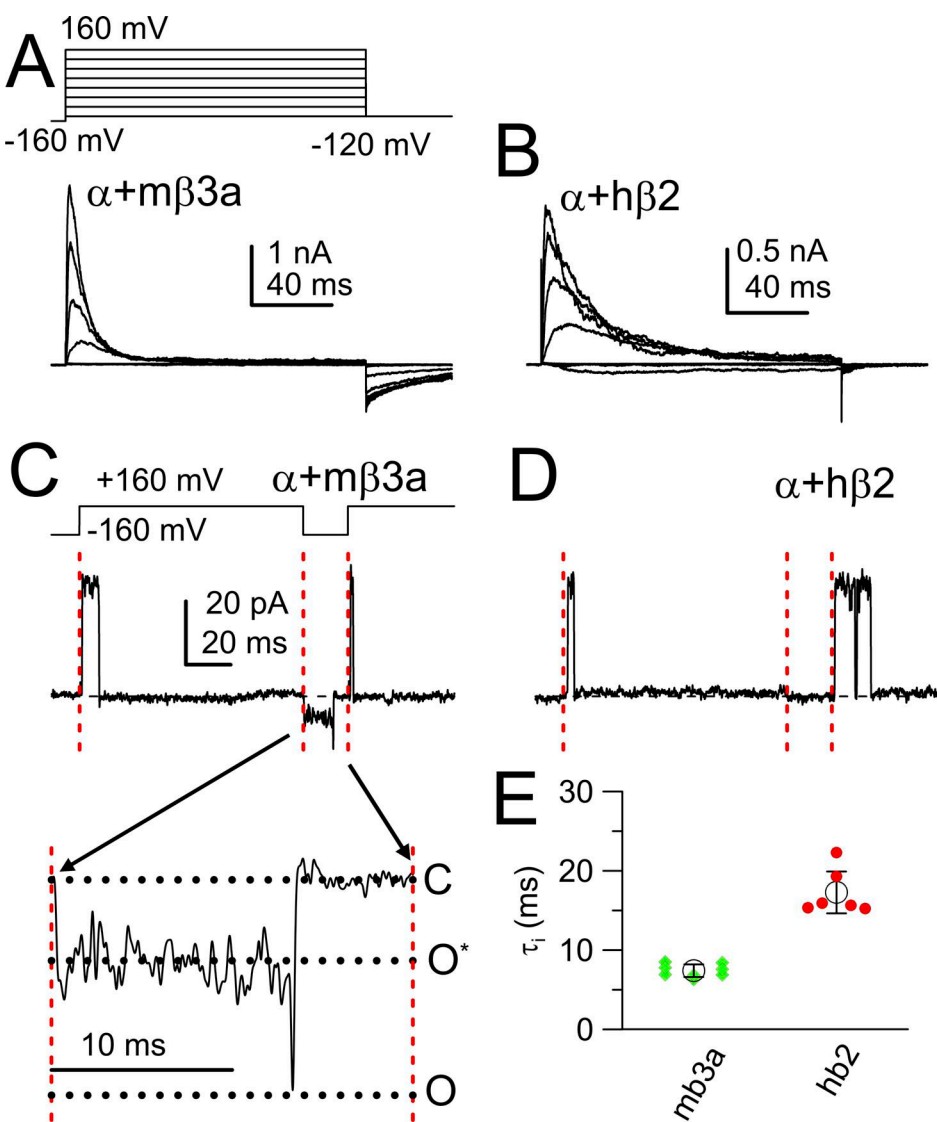

Figure 1. **Comparison of inactivation onset and tail current during recovery from inactivation for β2- and β3a-mediated inactivation at 10 μM [Ca²⁺]ᵢₙ. (A)** Inactivation mediated by mβ3a with protocol shown on the top. **(B)** Current inactivation with hβ2. **(C)** Trace shows a single α+β3a channel tested with the indicated protocol. The channel activates and inactivates upon depolarization, upon repolarization to −160 mV immediately reopens to an open level of reduced apparent conductance level, then opens to a fully open BK current level before closing. Finally, the channel reopens upon depolarization, indicative that it has recovered from inactivation. The inset highlights this behavior, displaying the entire current record between repolarization and subsequent depolarization, indicated by red vertical lines. Horizontal dotted lines indicate the 0-current level, subconductance level, and the amplitude of the unresolved final full opening. **(D)** Traces illustrate behavior of a single α+β2 channel, which inactivates but exhibits no tail opening upon repolarization, although the channel is available for opening with subsequent depolarization. **(E)** Mean + SD of inactivation time constants for 8 β3a patches and 6 β2 patches measured at +160 mV.

amplitude of the slow tail at −160 mV reflects an appreciable fraction (∼0.4) of the total maximal BK conductance in the patch measured at +160 mV (Gonzalez-Perez et al., 2012; Zhou et al., 2025).

Might estimates of the slow tail amplitude be compromised by changes in the slow tail time constant? To evaluate this, we measured α+β3a slow tail currents under two conditions that alter the relative average fraction of β3a IDs in the channel population. First, we measured slow tail time constants and amplitudes (normalized to maximum outward current at +160 mV after trypsin) for α+β3a currents arising from different α:β3a injection ratios ranging from 1:1 to 40:1 α:β3a (Fig. S1, A and B). Although the normalized slow tail amplitude decreased

by about 80% over this range of injection ratios, the slow time constants exhibited only modest changes (Fig. S1 B). Second, we measured slow time constants and normalized slow tail amplitude at different times of digestion by trypsin (Fig. S1, C and D). Again, over a large range of reductions in slow tail amplitude as the fraction of non-inactivating current increased, there was no appreciable change in the slow tail time constants (Fig. S1 D). This validates our use of slow tail amplitude as a measure of the fraction of channels inactivated by β3a at the end of depolarizations. It also supports the idea that the slow tails represent the behavior of a single β3a N terminus during recovery from inactivation, since reductions in the average number of N termini in a channel population do not alter the kinetics of that process.

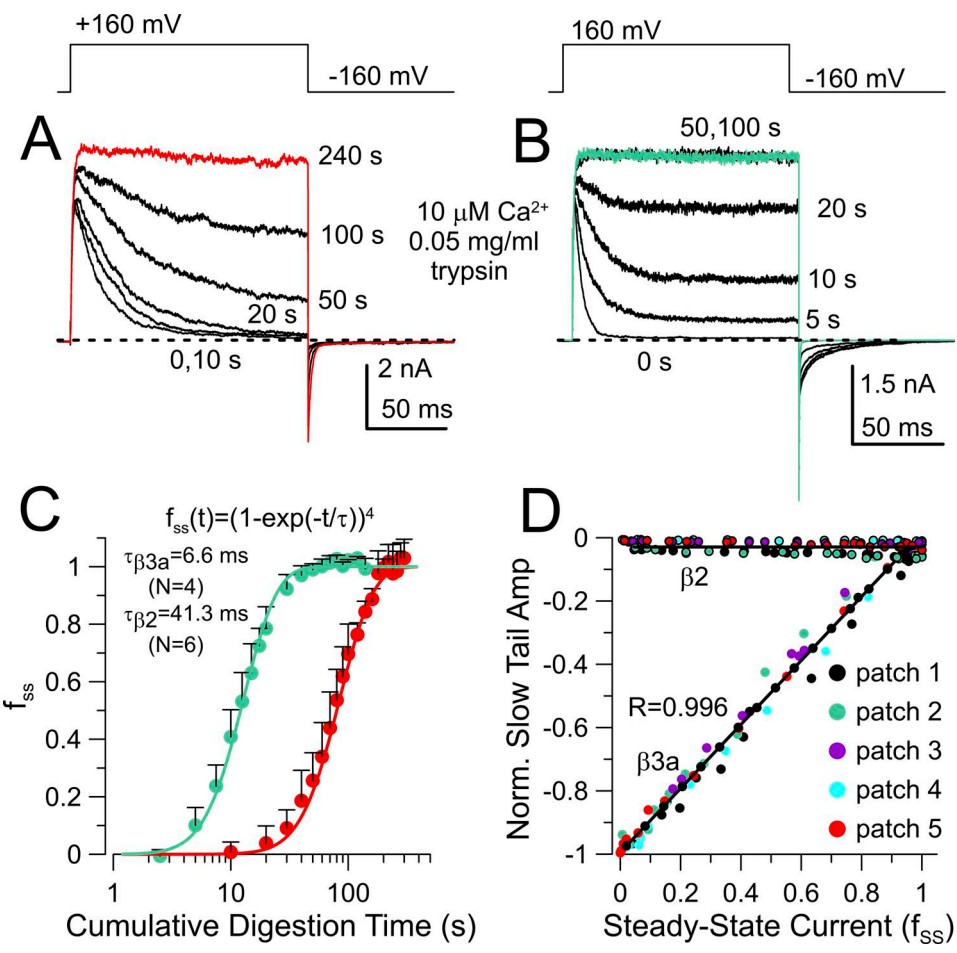

**Figure 2. Slow tail currents and rates of trypsin digestion distinguish between channels inactivated by β2 and β3a. (A)** BK+β2 currents at the indicated times of cumulative trypsin digestion. **(B)** BK+mβ3a currents during trypsin digestion. **(C)** Changes in steady-state current amplitude ($f_{ss}$) at different times of trypsin digestion. Means ± SD. Curves were fit with the indicated function, yielding the indicated time constants of digestion. N = number of patches. **(D)** Changes in normalized slow tail amplitude ($A_{ST}$) as a function of $f_{ss}$ measured at different times of trypsin digestion from 5 patches for β3a and 5 patches for β2. For BK+β3a, peak slow tail amplitude prior to trypsin was used to normalize subsequent amplitudes measured during trypsin digestion. For BK+ β2, initial tail current amplitude was measured at t = 3 ms after nominal repolarization, and minimal changes were observed with digestion.

**Approaches to evaluating α:β2:β3a subunit composition**

We hypothesized that, by taking advantage of slow tail currents and the differential rates of digestion of β2 and β3a N termini, this may allow tests that distinguish between the Segregated and random mixing models of assembly. Before turning to experimental tests involving β2 and β3a coexpression, we first present the theoretical expectations based on the two models. Do the two models make predictions that may prove to be experimentally distinguishable?

The underpinnings for the two models are given in detail in the Materials and methods. In brief, both models assume four beta-occupancy sites in a channel. The time course of removal of inactivation predicted by the segregated model is given by Eq. 11, which is derived from the sum of two binomially distributed populations (Eq. 8). In contrast, the random mixing model requires a trinomial distribution (Eq. 12) to take into account the occupancy of each site in individual channels by either β2, β3a, or an N terminus–removed β, resulting in a digestion time course defined by Eq. 14.

We first evaluate expectations that arise from coexpression of different ratios of β2:β3a subunits in regard to how slow tail

amplitude might be expected to vary simply as a function of mole fraction of β2 in the channel population. For the segregated model (Fig. 3 A), since we assume there is a sufficient quantity of β2 and β3a subunits to saturate available sites of β subunit occupancy, there are only the two stoichiometric possibilities, P(4β2,0β3a) and P(0β2,4β3a), such that, for the case illustrated (Fig. 3 A), the fraction of each combination in the population is simply the injected mole fractions, $p_{β2}$ = 0.5 and $p_{β3a}$ = 0.5. Since slow tail amplitude reflects exclusively those channels that become inactivated by a β3a N terminus, while β2-inactivated channels do not contribute at all to slow tails, the slow tail amplitude is predicted to diminish linearly with the increase in fraction of channels that are inactivated by β2 (Fig. 3 C). For the mixed model (Fig. 3 B), at any given β2:β3a injection ratio, the fractional occurrence of each potential stoichiometry, P(4β2,0β3a), P(3β2,1β3a), P(2β2,2β3a), P(1β2,3β3a), and P(0β2,4β3a) (Fig. 3 B), can be determined from the binomial distribution, since there are no sites lacking a β subunit. The predicted slow tail amplitude is calculated from the likelihood that a channel of each stoichiometry will be inactivated by β3a at

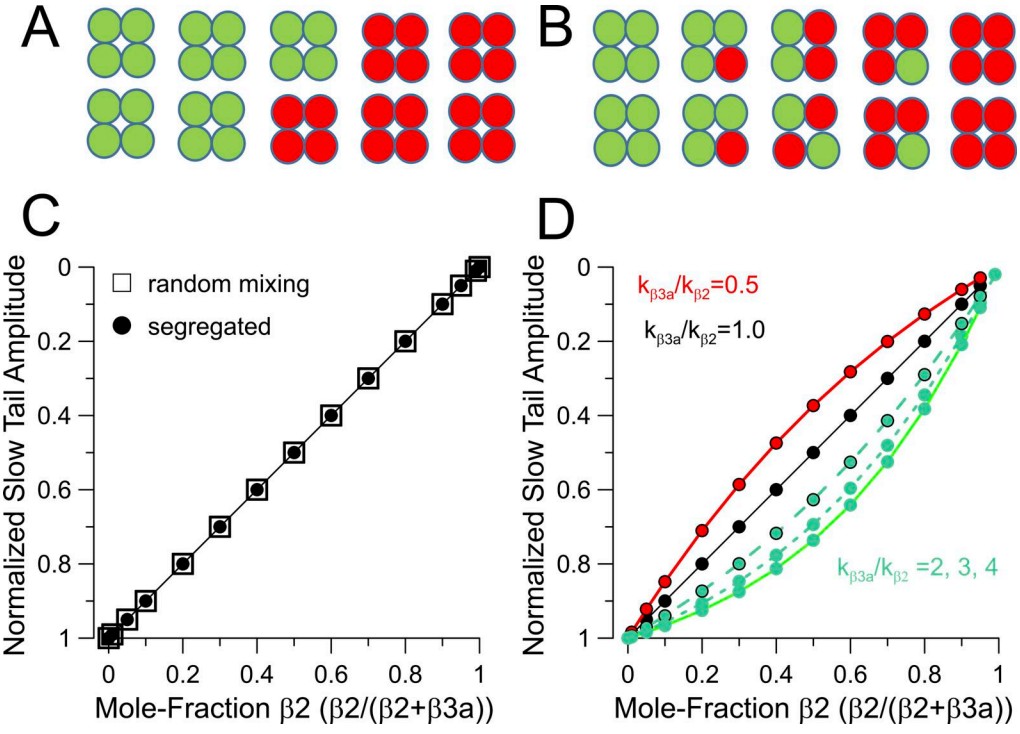

Figure 3. **Segregated vs. mixed models at full β subunit site occupancy. (A)** Schematic of how different β subunit isoforms would be distributed in a channel population in a segregated model. **(B)** Stoichiometric possibilities in accordance with a random mixing model. **(C)** Predictions for how slow tail amplitude ($A_{ST}$) will vary as mole fractions of β2 and β3a are varied and does not differ between random mixed and segregated models, assuming that likelihood of inactivation by either β2 or β3a N termini is identical. **(D)** Illustration of how differences in β2 and β3a inactivation likelihood will impact $A_{ST}$ as a function of nominal β2/β3 mole fraction, where $k_{β3a}/k_{β2}$ reflects the ratio of β2 and β3a likelihoods, e.g., as suggested by the ratio of differences in β3a vs β2 inactivation onset.

the end of a depolarization. For reasons of simplicity, we first assume that each individual N terminus, whether β2 or β3a, has an equal probability of producing inactivation, such that for a 3β2:1β3a channel, the likelihood that it ends up inactivated by β2 is threefold higher, and so on. With that assumption, the slow tail predictions for the mixed model result in a prediction for the relationship between normalized slow tail amplitude and mole fraction of β2 subunits that is identical to the segregated model (Fig. 3 C). This arises because, for any mole fraction injection ratio, irrespective of the model, there will always be the same fraction of β3a subunits in the channel population.

For a channel complex containing mixtures of β3a and β2 N termini, the assumption that any given N terminus has an equal likelihood of occupying the position of inactivation at the end of a depolarization is likely inaccurate. Based on the time constants of inactivation, the rate of inactivation for the β3a N terminus is at least threefold faster than for β2 (Fig. 1 C). However, differences in the lifetime of the inactivated states might also contribute to the likelihood of each type of subunit contributing to inactivation at the end of a depolarization, although we have no experimental evidence that addresses that possibility. Given the differences in rates of β2 and β3a-mediated inactivation, we have evaluated how such differences might impact on the expectations for the relationship between slow tail amplitude at a given β2/(β2+β3a) ratio. For two stoichiometries, P(4,0) and P(0,4), differences in inactivation likelihood can be ignored,

while for P(3,1), P(2,2), and P(1,3), we adjusted the likelihoods for inactivation by β3a relative to β2 as described in the Materials and methods. Taking into account the different inactivation rates, a curvature in the expected relationships between slow tail amplitude as a function of the β2/(β2+β3a) ratio (Fig. 3 D) is introduced. The potential impact of the differential likelihoods of inactivation on our tests of models of assembly will be revisited later. Irrespective of any adjustments required to account for differential inactivation likelihoods, the main point to be made here is that simple variation of injection ratios of two different subunit isoforms is not useful in distinguishing between assembly models.

**Manipulating channel stoichiometry by trypsin digestion**
The differences in trypsin digestion rates of β2 and β3a N termini allow the stoichiometric composition of individual BK channels to be altered as a function of cumulative time of trypsin digestion, with uniquely different predictions in accordance with either the segregated model (Fig. 4 A) or the random mixing model (Fig. 4 B). For the segregated model (Fig. 4 A), the distribution of a population of BK channels containing either β2 or β3a subunits or inactivation-removed subunits, as N termini are cleaved, follows expectations given by the sum of two binomially distributed populations. Predicted digestion curves plotting the fraction of non-inactivating current ($f_{ss}$) from different starting β2/(β2+β3a) ratios are described by the double exponential

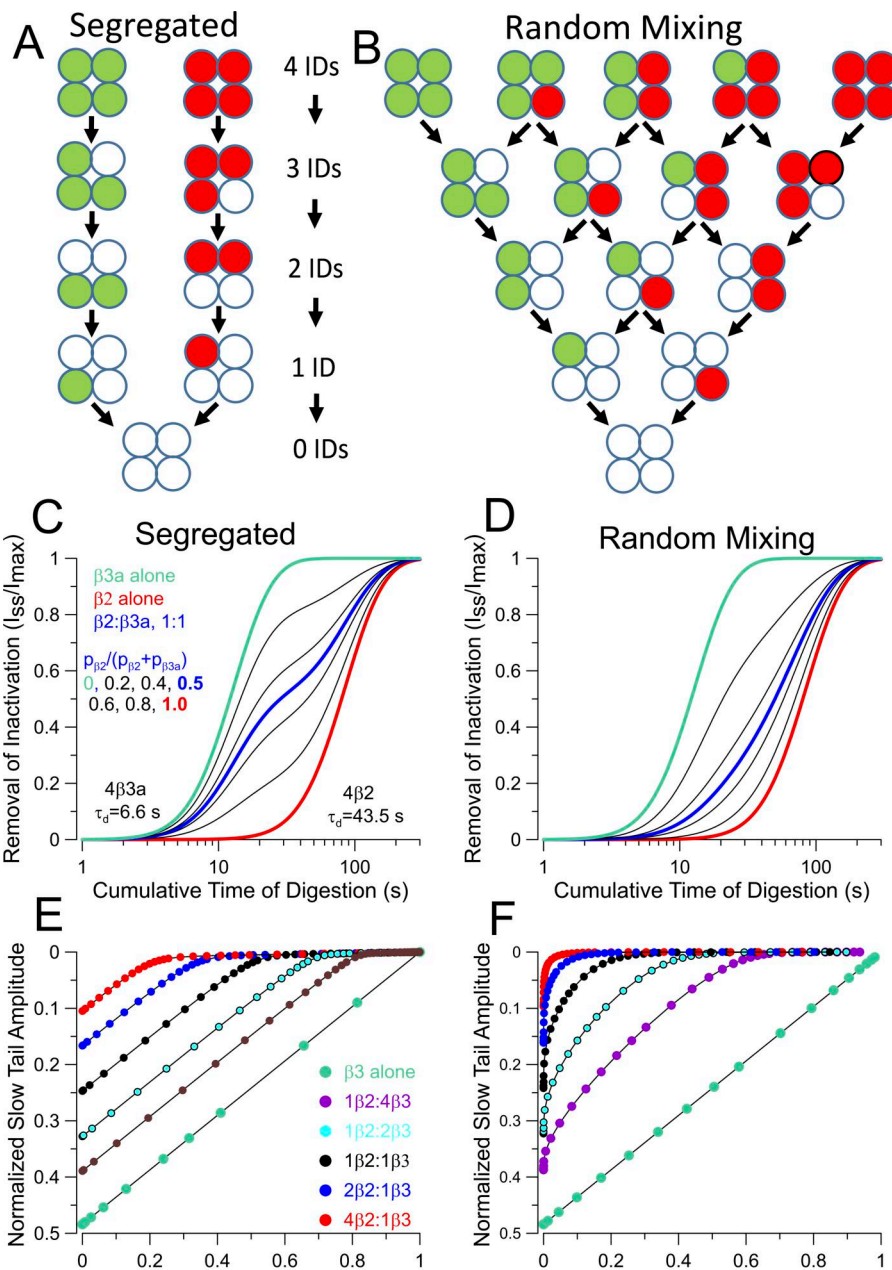

Figure 4. **Segregated and random mixing models make distinguishable predictions.** **(A)** Schematic of sequential digestion of N termini of β2 and β3a associated channels for assembly in accordance with segregated model. **(B)** Sequential digestion of N termini ($f_{ss}$) in accordance with random mixing assembly model. **(C)** Predicted time course of removal of inactivation for the segregated model for the indicated changes in the β2/(β3a+β2). **(D)** Predicted time course of removal of inactivation for random mixing model. **(E)** Relationship of fractional changes in slow tail current amplitude ($A_{ST}$) with fraction of non-inactivating current ($f_{ss}$) during trypsin-dependent removal of N termini for segregated model. **(F)** Tail current vs. $f_{ss}$ relationship for digestion in accordance with random mixing model.

given by Eq. 11 in the Materials and methods (Fig. 4 C). Notably, such curves exhibit a clear inflection point.

For the random mixing model (Fig. 4 B), the digestion results in 15 stoichiometric possibilities and requires application of the trinomial distribution analysis (see Materials and methods). We simulated the predicted time course of digestion in two ways: first, point-by-point calculation of the trinomial distribution expected based on changes in relative fractions of $p_{β2}$, $p_{β3a}$, and $p_{IR}$, i.e., the probability of occupancy by β2, β3a, or ID removed β's (IR), as $p_{β2}$ and $p_{β3a}$ are reduced in accordance with their separate trypsin digestion rates. From any initial starting distribution of occupancies by β2 and β3a subunits, P(β2,β3a), only when channels reach P(0,0) with inactivation fully removed do they contribute to $f_{ss}$. The predicted time course of changes in $f_{ss}$ is also analytically described by Eq. 14. A distinguishing feature

between the two models is the behavior at nominal 1:1 expression ratios, indicated by the green lines. Whereas for the segregated model, the curve is a simple double exponential function intermediate between the two extremes; for the mixed model, the curve is strongly shifted toward the β2 time course, despite the equal abundance of β3a in the population. This arises because of the pronounced slowing of digestion by the presence of even a single β2 subunit in a mixed channel.

The shapes of the predicted digestion curves (Fig. 4, C and D) appear sufficiently different that they may be amenable to experimental tests to distinguish between the models. These predicted changes in $f_{ss}$ depend in no way on the differences in slow tail properties between β2- and β3a-mediated inactivation and are only defined by the relative digestion rates of the two different IDs. As such, the issue raised above regarding the

differential likelihood that a channel may be more likely to be inactivated by a β3a ID than a β2 ID at the end of a depolarizing step does not apply.

Each model also makes different predictions in regard to how the slow tail current amplitude will vary as IDs are removed. As mentioned earlier, when only β3a is expressed, the two models are identical. However, as the fraction of β2 is increased, the initial normalized slow tail amplitude is reduced for both models because of the increase in the fraction of channels that are inactivated by β2 (Fig. 4, E and F). Moreover, the segregated model predicts that the changes in slow tail amplitude ($A_{ST}$) as a function of $f_{ss}$ is largely linear as digestion proceeds, consistent with the behavior of BK channels containing only β3a. At the most elevated fractions of β2, there is a wide range of changes in $f_{ss}$ for which there is no longer any slow tail current, reflecting the slow removal of inactivation of primarily β2-containing BK channels (Fig. 4 E). In contrast, for the random mixing model, at any ratio of both β2 and β3a coexpression, changes in $A_{ST}$ exhibit marked curvature as IDs are removed. Since these plots are influenced by the properties of the slow tails, we also evaluated the impact of changes in the likelihood of inactivation ($L_{β3a}$ vs. $L_{β2}$) at the end of a depolarization for individual β3a vs. β2 N termini. For the segregated model, since no channels contain both β2 and β3a subunits, any difference in relative β2 and β3a inactivation likelihoods is irrelevant. For the random mixing model, predicted $A_{ST}$ vs. $f_{ss}$ curves during digestion were generated for $L_{β3a}/L_{β2}$ ratios of 0.5, 1, 2, 3, and 4 (Fig. S2), with ratios of 3 or 4 being most consistent with the measured differences in inactivation time constants (Fig. 1 C). The differences in the idealized curves obtained for $k_{β3a}/k_{β2}$ of either 1 or 4, although clear (Fig. S2, E and F), are less than observed between the segregated and random mixing models.

For the considerations above, we have assumed that, irrespective of the model, all sites for β subunit occupancy are fully occupied, whether by β2 or β3a subunits. Earlier work, both from native cells (Ding et al., 1998) and with heterologous expression (Wang et al., 2002), indicates that BK channel complexes with <4 β2 subunits can occur. Although the experimentally determined trypsin digestion time courses for β2 and β3a removal of inactivation (Fig. 2) support the idea that almost complete occupancy by β subunits occurs in the present experiments, we were concerned that the occurrence of partial occupancies might compromise our ability to distinguish usefully between segregated and random mixing models. We therefore evaluated the predictions for a segregated model with the assumption that, once one type of β subunit associates in a complex, only other identical subunits can assemble. However, partial occupancy within the segregated populations can occur. Within this framework, we compared the impact of occupancies of 1.0, 0.8, and 0.6 within the segregated β2 and β3a channel populations (Fig. S3). With partial occupancy of 0.6, the fit of the predicted trypsin digestion time course for either the β2 alone population or the β3a alone population requires exponents of <2, clearly less than what is observed experimentally here (Fig. S3 C). However, even if partial occupancy were 0.6, the predicted dependence of slow tail amplitude as a function of the steady-state current at different time points of trypsin digestion would

still mirror the linear relationships for the segregated model assuming full subunit occupancies (compare Fig. 4, E and F; and Fig. S3, D and E. Thus, partial occupancy in segregated populations does not predict behaviors that would be consistent with the random mixing model.

## Time course of digestion by trypsin supports a random mixing model

Guided by the above considerations, we expressed different ratios of β2 and β3a messages with mSlo1 and hβ2 at a fixed 1:1 weight ratio, with β3a added in to define the nominal β2:β3a ratios. Currents in excised patches were then recorded at different cumulative times of application of 0.05 mg/ml trypsin (Fig. 5, A–F). Patches were bathed with 0 Ca²⁺ during application of trypsin, but sample currents monitoring trypsin digestion were obtained with 10 µM Ca²⁺. In all cases, patches were only used if full removal of inactivation was achieved. As expected, steady-state current measured at the end of the depolarization to +160 mV increases more rapidly with trypsin digestion of patches with a larger fractional expression of β3a. In addition, it will be noted that all the experimentally injected ratios correspond to nominal mole fractions of β3a message much in excess of β2, all being at least 0.8 and higher. Even at the lowest nominal mole fraction of β3a, 0.8 (1β2:4β3a; Fig. 5 B), which would imply a fourfold excess of β3a relative to β2, the slow tail current amplitude remains only about half of that for β3a alone. Despite the large excess of β3a available, slightly less than half the channel population apparently has been inactivated by β3a. Based on the considerations above (Fig. 4 E), a slow tail amplitude that is half that of β3a alone would be expected to occur at 1:1 β2:β3a. This suggests that, under the conditions of our experiments, channel assembly, irrespective of the assembly model, is not occurring in accordance with the nominal mole fractions given by the injected subunit ratios.

From such current records, we measured the fraction of non-inactivating current at the end of each depolarization normalized to that after complete digestion ($f_{ss}$) and also determined the amplitude of the slow tail normalized to current at +160 mV following removal of inactivation ($A_{ST}$). Plots of the $f_{ss}$ as function of digestion time yielded digestion time courses intermediate between that of BK+β3a alone and that of BK+β2 alone (Fig. 6, A and B). For the random mixing model, all digestion curves were initially fit simultaneously to the following variation of Eq. 14 (see Materials and methods):

$$I(t) = \left(1 - p_{β2}(0)e^{-\frac{t}{τ_{β2}}} - p_{β3a}(0)e^{-\frac{t}{τ_{β3a}}}\right)^n, \quad (14a)$$

where $p_{β2}(0)$ and $p_{β3a}(0)$ represent starting mole fractions of β2 and β3a, respectively, for a given digestion time course, $τ_{β2}$ and $τ_{β3a}$ are the digestion times constants, and n is the number of IDs per channel. With the nominal values of $p_{β2}$ and $p_{β3a}$ based on the injected cRNA ratios, no convergence was achieved. However, if individual curves were fit to the above equation, leaving $p_{β2}$ as a free parameter, with $p_{β3a} = 1-p_{β2}$, adequate fits were achieved, but with nominal $p_{β2}$, $p_{β3a}$ values quite different than expected based on injected ratios. We therefore hypothesized that whatever the basis for the different

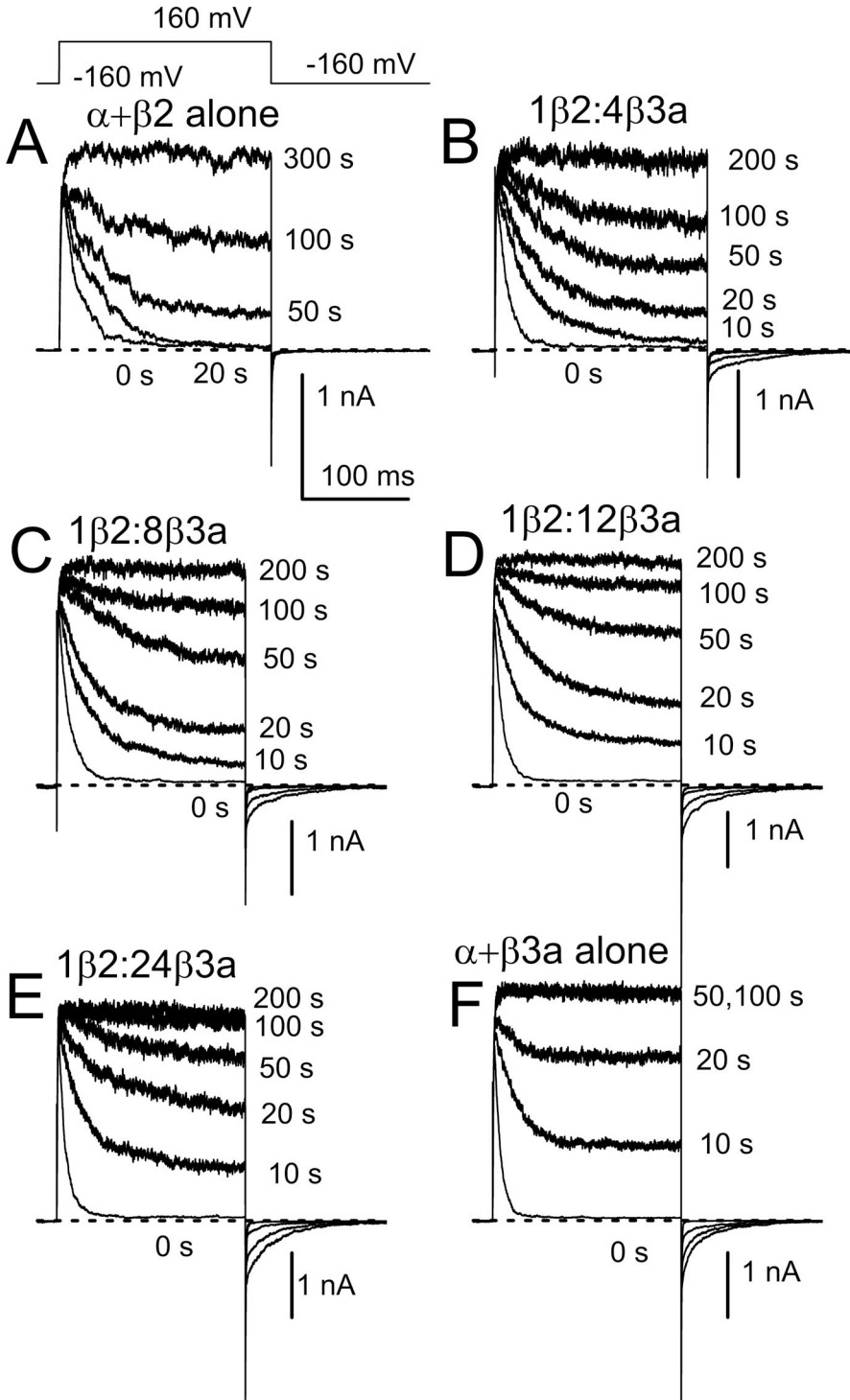

Figure 5. **Trypsin-mediated removal of inactivation and slow tails for BK currents arising from different β2:β3a injection ratios.** **(A)** Currents arising from BK α+hβ2 alone with each trace at the indicated cumulative time of trypsin application (0.05 mg/ml). Currents were activated with 10 μM Ca²⁺. **(B)** Currents with nominal 1β2:4β3a. **(C)** 1β2:8β3a. **(D)** 1β2:12β3a. **(E)** 1β2:24β3a. **(F)** α+β3a alone.

effectiveness of expression or assembly of translated product might be, it should apply equivalently to all injected β2/β3a ratios. For nominal injected amounts of β2 and β3a, say, $Q_{β2}$ and $Q_{β3a}$, the nominal mole fractions of β2, $p_{β2} = Q_{β2}/(Q_{β2}+Q_{β3a})$, with $p_{β3a} = 1-p_{β2}$. We hypothesize that the effective activity of injected β2 is increased by a constant scale factor, $sf$, relative to that for β3a. Thus, we define the effective β2 injected amount as $Q_{β2}' = sf*Q_{β2}$, with the effective $p_{β2}' = sf*Q_{β2}/(sf*Q_{β2}+Q_{β3a})$. Thus, for a nominal β2:β3a injection ratio of 1:4 (corresponding to $p_{β2} = 0.2$),

for $sf = 5$, the effective injection ratio becomes 5:9 and the effective β2 mole fraction is $p_{β2}' = 0.55556$. With $sf$ as an additional free parameter, the resulting equation used for simultaneous fitting of the digestion curves becomes

$$I(t) = \left(1 - p_{β2}'(0)e^{-t/τ_{β2}} - p_{β3a}(0)e^{-t/τ_{β3a}}\right)^n. \quad (15)$$

To evaluate the adequacy of the two models, for the random mixing model, we fit the family of curves with Eq. 15, while for the segregated model we used Eq. 5. Irrespective of the fitting

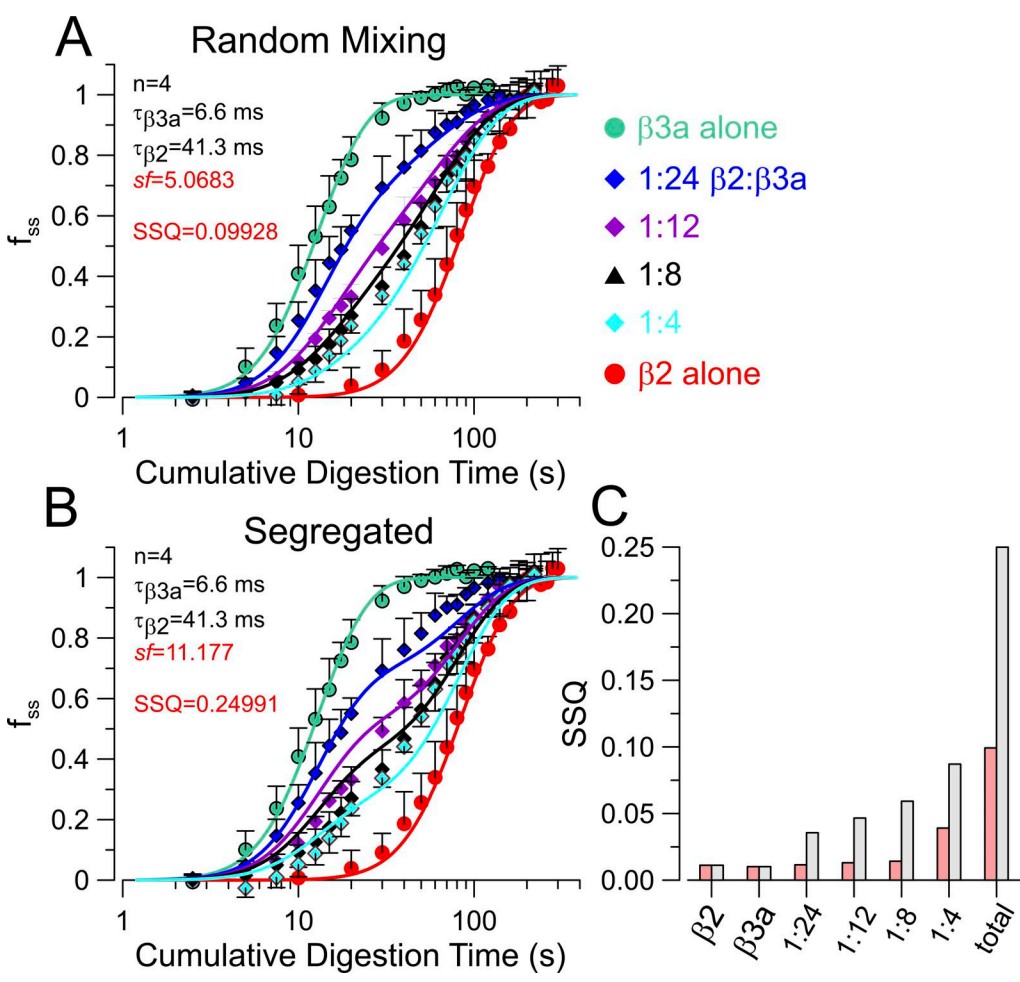

Figure 6. **The random mixing model better describes trypsin digestion time course than the segregated model. (A)** Normalized digestion time course for the indicated (right) nominal injection ratios. Lines are best fits of Eq. 15 (random mixing), with all curves fit simultaneously and only *sf* was a free parameter (red), while parameters in black were constrained to the indicated values. **(B)** Simultaneous fit of the segregated model. **(C)** Sum of squares (SSQ) for comparisons of curves to each individual time course, and also all values together, with the random mixing yielding a fit about 2.5-fold better with the same number of free parameters. Red: random mixing; Gray: segregated. The fit parameter, n, refers to the average number of inactivation domains per BK channel in the channel population.

constraints (Fig. S4), the random mixing model yields a sum of the squares 2.5-fold lower than that for the segregated model. Constraining the digestion time constants to the values estimated from fitting the trypsin digestion time course for β3a alone and β2 alone and assuming all channels begin with $n = 4$ IDs, the full set of digestion time courses at different injection ratios can be reasonably well fit with a single free parameter ($sf$) (Fig. 6), yielding an SSQ of 0.0993, not too much different from the SSQ = 0.0929 obtained with 4 free parameters. That the scale factor constant, $sf$, works across all four injection ratios gives support to the idea that it reflects an intrinsic difference in efficacy of the β2 message or protein expression process in relation to β3a. Whatever the origins of this β2/β3 difference, the key point is that the mixed model better accounts for the observations.

**Relationship between slow tail and $f_{ss}$ supports random mixing**

The analysis above utilized only the time course of changes in $f_{ss}$ as a function of trypsin digestion for different digestion ratios,

whereas we showed above that the two models also exhibit different predictions for the behavior of slow tail amplitude as a function of changes in $f_{ss}$. Although we do not have an analytic equation that permits fitting of the relationship between $A_{ST}$ and $f_{ss}$ for the random mixing model, we compared the predictions for the $A_{ST}$ and $f_{ss}$ the relationship predicted based on the fitted values for the random mixing and segregated models (Fig. 7). For each set of patches evaluated with a given β2:β3a injection ratio, $f_{ss}$ and $A_{ST}$ values were separately averaged for each time point of trypsin application, resulting in mean ± SD values for both $f_{ss}$ and the associated slow tail amplitude ($A_{ST}$, normalized to the current at +160 mV after trypsin digestion) (Fig. 7 A). From the fitted values obtained from the analysis above (Fig. 6), any given $f_{ss}$ is predicted to define $A_{ST}$ for each of the two models. However, rather than simply normalize to the maximum $A_{ST}$ for the β3a alone data, we chose to step through a set of $A_{ST}$ values (0.2 to 0.25) bracketing those obtained with β3a alone. This identified the $A_{ST}$ value that yielded the minimum SSQ for the two cases: random mixing (Fig. 7 B) and segregated (Fig. 7 C). The

SSQ for random mixing was almost half of that for the segregated model. It should also be noted that the random mixing model better recapitulates the curvature predicted in the $A_{ST}$ vs. $f_{ss}$ relationship than the segregated model (also consider Fig. S3).

## β2 and β3a coassemble in single BK channels

We collected single-channel recordings from oocytes expressing BK α together with either β2 or β3a separately and then from patches injected with BK α with a 1:8 β2:β3a injection ratio. Patches were stimulated with a 100 ms depolarization to +160 mV, followed by a 20 ms repolarization to –160 mV and then followed by a second step to +160 mV (Fig. 8). Channel activity was then evaluated for whether the channel opened and inactivated during the initial depolarization, whether there was a tail opening following repolarization, and then whether the channel was again available for activation upon the second depolarization. For β2-containing channels, repolarization was invariably followed by an absence of openings, although more often than not the subsequent depolarization clearly revealed the channel had recovered from inactivation (Fig. 8 A). In contrast, for β3a-containing BK channels (Fig. 8 C), repolarization usually resulted into immediate opening in the O*-I subconductance behavior, which in some cases was terminated by an opening to a full BK tail opening level. Subsequent depolarization revealed that in many cases the β3a-containing channel may have recovered from inactivation, although in cases where a channel remained in the O*-I behavior at the end of the 20 ms repolarization, subsequent depolarization generally failed to produce opening. With oocytes in which β2 and β3a were coexpressed, we obtained 16 patches that exhibited β3a-type tail openings, but 14 of these patches also exhibited repolarization intervals where no opening was observed, but which were generally followed by a subsequent opening during the second depolarization, a characteristic of the β2-containing BK channels. However, such behavior can also be seen, albeit more rarely, for some patches expressing only β3a (trace 3 in Fig. 8 C), so it is not definitively predictive of either β2- or β3a-mediated inactivation. Yet, the presence of the instantaneous reduced amplitude tail openings is only associated with inactivation mediated by a β3a N terminus. As such, we determined the frequency of occurrence of such β3a-type tails for all sweeps for a set of 5 patches from β2 alone oocytes, 6 patches from β3a alone oocytes, and 16 patches with 1:8 β2:β3a coexpression (Fig. 8 D). β3a-type tails were never observed in β2-patches, at a frequency in excess of 0.9 for β3a-patches, and with a broad distribution of frequencies in patches with coexpressed β2:β3a subunits. In principle, the random assembly model predicts four potential stoichiometries containing at least one β3 a subunit (4:0, 3:1, 2:2, and 1:3 β3a:β2). 15 of 16 patches from coexpression had at least 40 trials, but one had only 14. Overall, the range of values is consistent with there being multiple combinations, but the number of patches and probably the number of sweeps for each patch are insufficient to look for groupings.

For the simplest case, where both β2 and β3a N termini may exhibit similar likelihoods of producing inactivation at the end of a depolarization ($L_{β2} = L_{β3a}$), one would expect groups at 1.0,

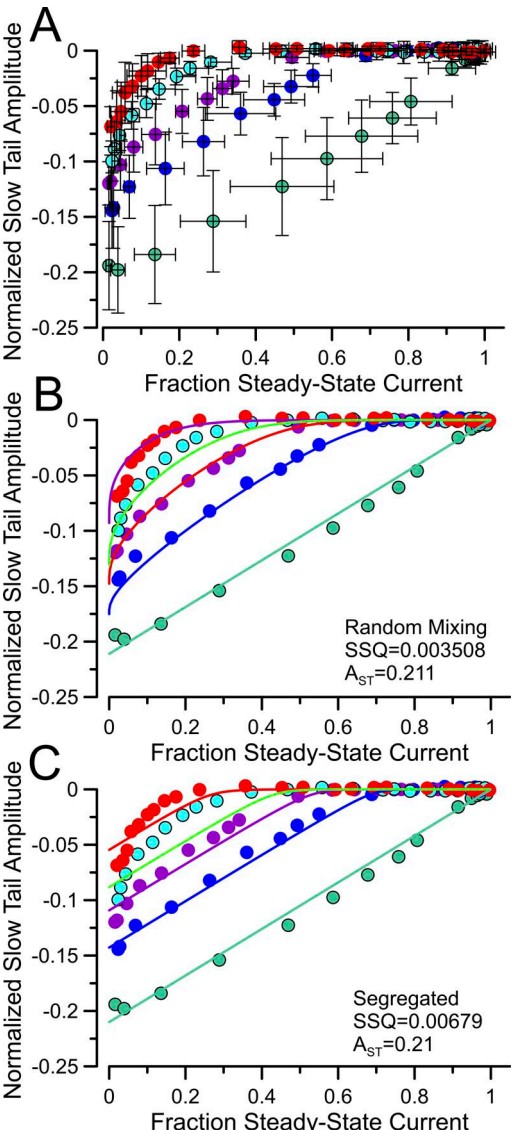

Figure 7. **Changes in fractional slow tail amplitude with changes in $f_{ss}$ also support random mixing. (A)** For each injection ratio, $f_{ss}$ and tail current amplitudes normalized to $BK_{Gmax}$ for a patch ($A_{ST}$) were measured at different times of removal of inactivation by trypsin. Resulting means and SD for both $f_{ss}$ and $A_{ST}$ values are plotted. **(B)** Based on the fitted parameters for the random mixing model from Fig. 6, fractional slow tail was calculated for each $f_{ss}$ value and overlaid over the values from A. The maximal $A_{ST}$ value for the idealized curves was stepped between 0.20 and 0.25 to yield the minimum SSQ (0.003508) between experimental and idealized. **(C)** Based on fitted values for the segregated model, idealized curves for $A_{ST}$ as a function of $f_{ss}$ were generated and overlaid over the experimental data. The minimal SSQ (0.00679) was obtained at $A_{ST} = 0.21$. SSQ, sum of squares.

0.75, 0.5, and 0.25 fractional likelihoods for 4:0, 3:1, 2:2, and 1:3 β3a:β2 assemblies. Qualitatively, the spread of observed values appears skewed to higher likelihoods of β3a tails (Fig. 8 D). As noted above, differences in inactivation rates between β2 and β3a (note ensemble average time constants; Fig. 8, A–C) suggest that the likelihood of being inactivated at the end of the 100 ms depolarization is higher for β3a than for β2. Although our results are insufficient to make any definitive interpretations regarding stoichiometry, we ranked our observations and

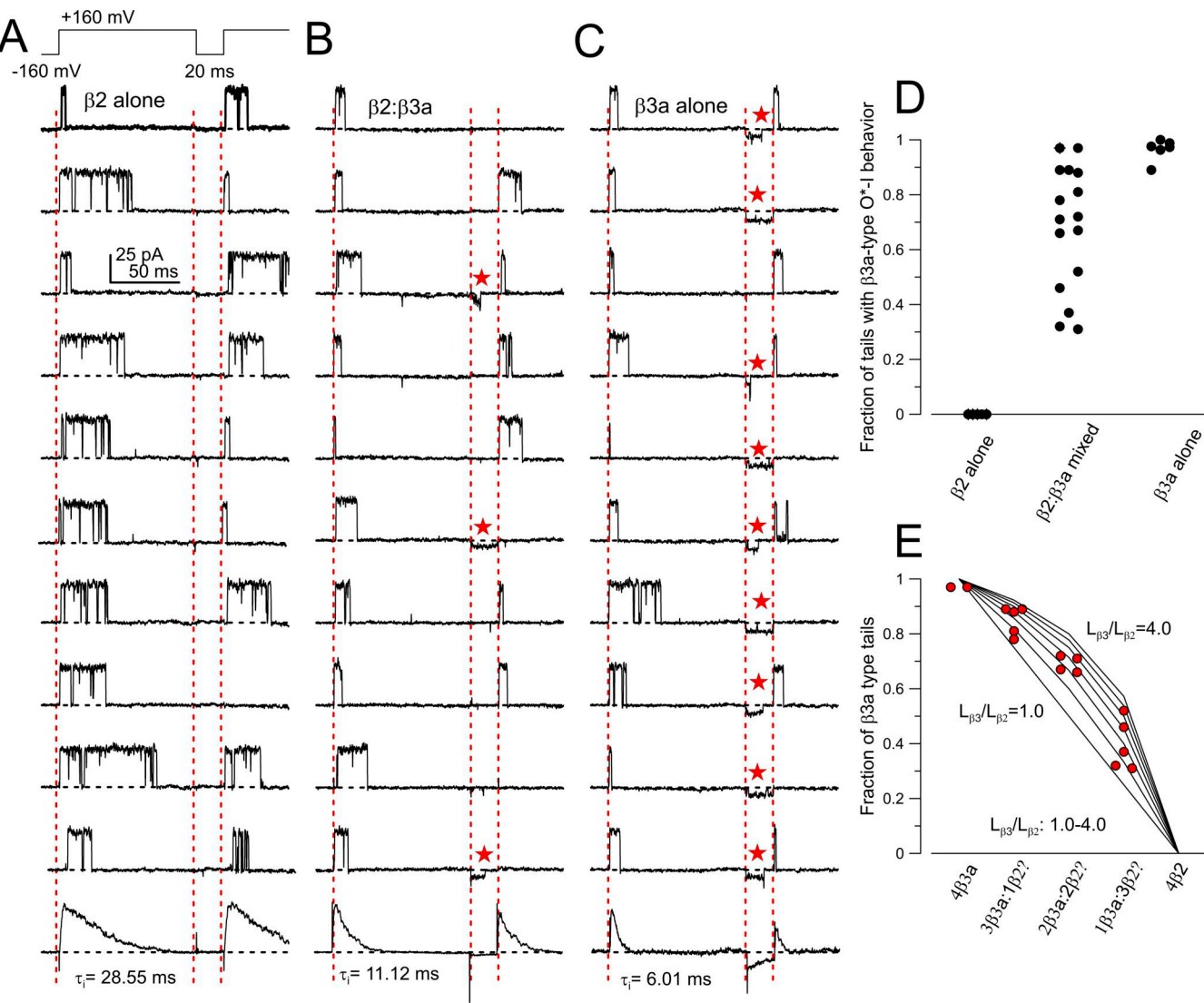

Figure 8. **β2 and β3a subunits coassemble in single BK channels. (A)** The indicated voltage protocol was used to activate single channels in excised patches (10 µM cytosolic Ca²⁺). Traces are from one patch with a single β2-containing BK channel, with ensemble average of all sweeps from the patch shown on bottom. For these 10 trials, 10 openings result in inactivation, but in 9 of 10 trials, the channel recovers from inactivation without reopening during the 20 ms repolarization to –160 mV. **(B)** Single-channel traces from a patch from an oocyte injected with a 1:8 β2:β3a injection ratio. 3 of 10 trials exhibit an instantaneous reopening upon repolarization with a reduced current amplitude (β3a-like; red stars) compared with a full BK tail opening. 5 of 10 trials show recovery from inactivation without any reopening during the repolarization (β2-like). **(C)** For a patch expressing a single β3a-containing BK channel, 9 of 10 traces show instantaneous reopening upon repolarization, with the reduced current amplitude relative to a full BK tail opening, The 5th and 10th traces show a brief full BK opening at the end of the reduced current burst. **(D)** Likelihood of observing β3a reduced conductance tail openings over all trials is plotted for 5 patches from β2-only patches, 16 patches from 1:8 β2:β3a injection, and 6 patches from β3a-only. **(E)** Red symbols plot likelihood of β3a-tail openings for the channels from the 16 β2:β3a patches, arranged in accordance with 5 potential stoichiometric assemblies: 4:0, 3:1, 2:2, 1:3, and 0:4 (β3a:β2). Lines plot predictions based on different ratios of likelihood (from 1.0 to 4.0) that a β3a N terminus produces inactivation at the time of repolarization relative to a β2 N terminus, assuming that this is defined solely by differences in rate of inactivation onset.

placed these over the predictions for cases in which $L_{β3a}/L_{β2}$ varies from 1 to 4 (Fig. 8 E). The measured values are fully consistent with the existence of four possible β2/β3a stoichiometries, with $L_{β3a}/L_{β2}$ being about 2–3. It should be noted that no β2 alone-type channels were observed in the 16 single channels from the coinjection oocytes. Overall, we can conclude unambiguously that both β2 and β3a subunits can coassemble in single BK channels, and the basic inactivation behavior of each of the subunits does not seem altered by the presence of the other isoform.

**Biochemical experiments report β2:β3a:α ternary complexes and intrinsic differences in β2 and β3a assembly likelihood**

We co-transfected suspension-adapted HEK293F cells with hSlo1_EM-TS, β2-RF and β3-Gρ (where TS, RF, and Gρ indicate twinstrep, mCherry-FLAG, and eGFP-ρ1D4 tags, respectively, appended onto the C terminus of the expression constructs), using plasmid weight ratios of 1:1:1 or 1:0.2:1.8. These two cases of transfection will be referred to as TR1 and TR2, respectively. Total protein was extracted in digitonin micelles, fractionated using a three-step purification scheme (Fig. 9 A), and each

fraction was analyzed with dual-color fluorescence size exclusion chromatography, using eGFP and mCherry fluorescence (Fig. 9 B). In total cellular lysates, based on calibrated fluorescence intensities, the molar ratios of β2-RF:β3-Gρ were found to be ∼ 3.3 ± 0.21 (TR1) and ∼ 0.75 ± 0.18 (TR2) (Fig. 9 C). After the first affinity purification step, which isolates total Slo1, the ratio of eGFP to mCherry fluorescence intensities at the peak retention volume of the chromatogram (ρGC) indicated the presence of β2-RF and β3-Gρ, affiliated with Slo1, in a molar ratio of 7.32 ± 0.48 (TR1) and 1.08 ± 0.16 (TR2) (Fig. 9 C). These values are informed by the relative distributions of the binary (Slo1:β2 and Slo1:β3) and ternary (Slo1:β2:β3) complexes. Two additional sequential affinity purification steps capture Slo1 complexes containing both β2 and β3 subunits. For these triply purified proteins, the FSEC profiles showed robust eGFP and mCherry fluorescence, and ρGC indicated the presence of β2-RF and β3-Gρ in a molar ratio of 2.33 ± 0.11 (TR1) and 0.89 ± 0.09 (TR2) (Fig. 9 C). These reflect ternary complexes, Slo1:β2:β3, of mean stoichiometry 4:2.8:1.2 and 4:1.9:2.1, respectively. The nonintegral stoichiometry of the subunits indicates that the ternary complexes are most likely a heterogenous mix of complexes of different stoichiometries.

Comparison of the peak fluorescence intensities in the input and flow-through fractions of the last affinity capture step also allowed us to estimate the relative distribution of the binary and ternary complexes (Fig. 9 D). As would be expected, the FSEC profiles of the isolated binary complexes exhibited only a single fluorescence signal (Fig. 9 B), distinguishing them from the ternary or the heterogenous mix of complexes in other fractions. Ternary complexes were found to comprise 14 ± 4.1% or 58 ± 6.2% of the total Slo1–β2 complexes in the TR1 and TR2 cases, respectively, the remainder being binary Slo1–β2 complexes. Similarly, we estimated that ternary complexes comprised 49 ± 4.4% and 37 ± 6.6% of Slo1–β3 complexes in the instances of TR1 and TR2. The relative proportion of the ternary complexes inferred from our biochemistry experiments differs from that of the electrophysiological results. This perhaps originates from the use of different expression constructs or systems. Additionally, while electrophysiological methods reflect plasma membrane complexes, the biochemical methods inspect the pool of total (digitonin-extractable) protein. Nevertheless, consistent with electrophysiological results, the biochemistry experiments indicate that under both co-transfection conditions (TR1 and TR2), both binary and ternary complexes co-exist, that ternary complexes comprise a substantial proportion of the total pool of protein complexes, and that the relative proportion of the binary and ternary complexes and the stoichiometry of the ternary complexes vary with the expression levels of the auxiliary subunits.

## Discussion

Through a combination of analysis of macroscopic current, single channels, and biochemical tests, the results show unambiguously that β2 and β3a subunits of BK channels can coassemble in the same individual BK channel complexes. Taking advantage of differences in the time course of trypsin digestion of β2 and β3a N termini and also differences in β2- and β3a-mediated tail current properties, we measured tail current amplitudes and steady-state current amplitudes at different cumulative durations of trypsin application. Coupled with the application of a trinomial distribution analysis, we explicitly evaluated two models by which β subunits might be assembled in BK channel complexes: first, a random mixing model in which the two distinct β subunits can independently become associated with any one of the four total β subunit–binding sites in the BK channel tetramer, and second, a segregated model, in which BK channels can only contain up to four of a single species of BK β subunit. The random mixing model provided the best fit over four tested β2:β3a injection ratios. However, irrespective of the model, the optimal fit obtained by either model required introduction of a scaling factor (sf), which scaled the nominal injected mole fractions of β2 and β3a subunits to adjusted mole fractions. This presumably reflects some intrinsic difference between β2 and/or β3a message, protein synthesis, and/or capacity for assembly resulting in an effective β2 availability being about sixfold greater than that of β3a. The potential underpinnings of the requirement for this sf are addressed below.

The random mixing model provided not only the better fit to the overall data but also better captured the nature of the curvature in the both the digestion time course and also the relationship between tail current amplitude and $f_{ss}$. One assumption in our analysis is that all potential sites of β subunit occupancy are filled, whether by β2 or β3a. Although the molar amounts of β2 and/or β3a messages in our experiments are expected to be near saturation, there are suggestions of less than full stoichiometric assembly in some experiments, e.g., even with 4:1 β2:α. For example, the best fit of the trypsin digestion time course for either β2 or β3a can result in best fits with power terms somewhat <4.0. However, our evaluation of predicted power terms suggests that, under the conditions of our experiments, the observed power terms are consistent with both β2 and β3a fractional occupancies, in the worst case being in excess of 0.9 (power term of ∼3.4). We were concerned whether it was possible that segregated assembly might be occurring, but that less than full stoichiometric assembly might alter our digestion time course data to mask the predicted inflection points, making a segregated model appear like a random mixing model. However, our evaluations excluded that possibility.

Whereas the single-channel and biochemical tests provide direct support that coassembly does occur, such experiments are less useful for discerning particular models of assembly. For single channels, the challenge is that it is unrealistic to obtain enough separate patches under various coexpression conditions to determine likelihoods of various stoichiometries quantitatively, while for biochemistry the problem is that the extent of transfection may differ among cells, and furthermore, the purified protein does not exclusively reflect surface membrane channels. As such, although the macroscopic experiments are more indirect in regard to supporting coassembly, to the extent they do support coassembly, they suggest that, at least β2 and β3a, coassembly in BK complexes is guided by independent random assembly of available subunits. However, this may not be the case for other BK β subunits, and it should be noted that

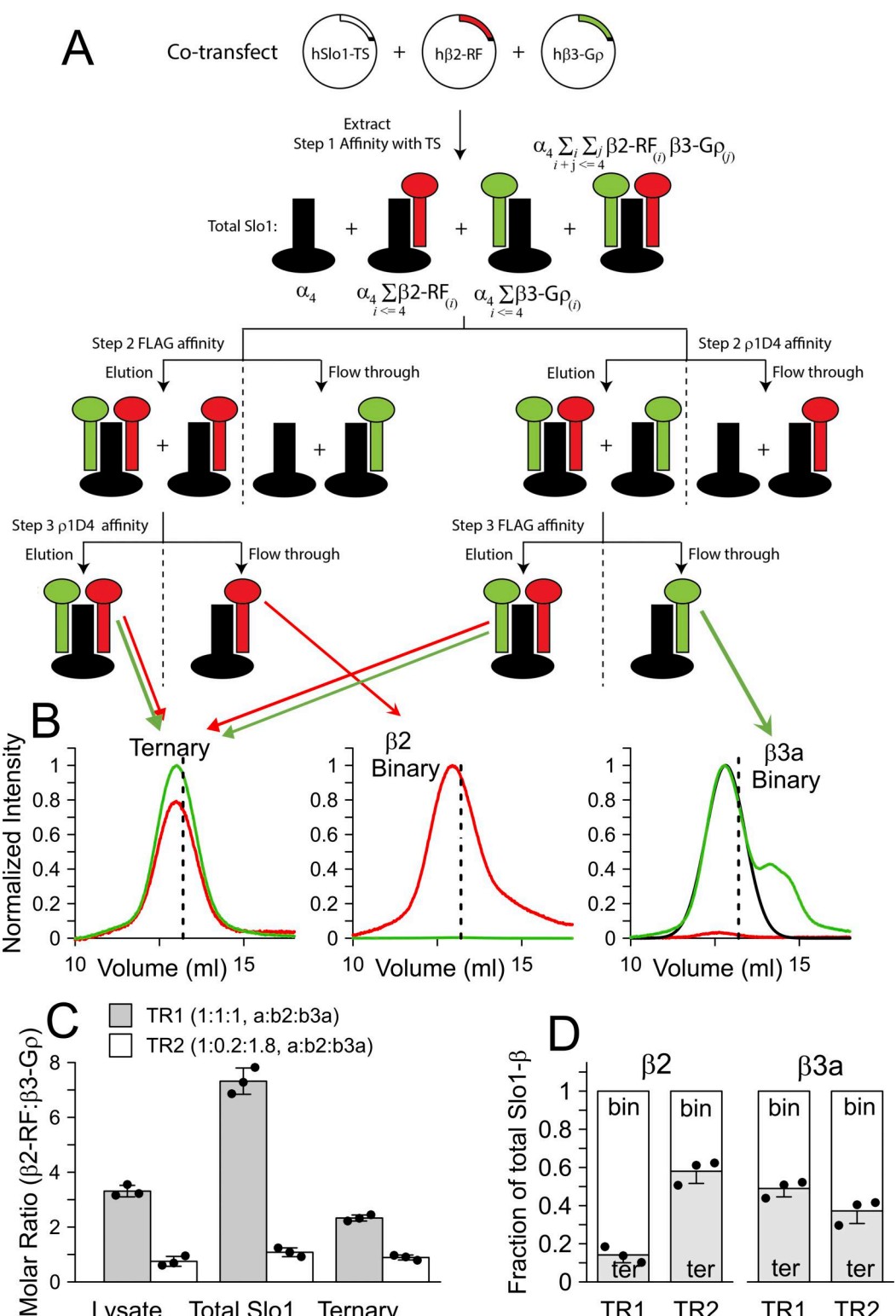

Figure 9. **Biochemical tests demonstrate presence of both binary and tertiary complexes with β2 abundance in complexes more likely than β3a at equimolar initial expression. (A)** 3-step affinity purification scheme to fractionate coexpressed Slo1, β2, and β3 into binary (Slo1–β2 or Slo1–β3) and ternary (Slo1–β2–β3) complexes. **(B)** Dual-color FSEC chromatograms of three key fractions (as indicated in A, via arrows) monitored via eGFP (green trace) and mCherry (red trace) fluorescence. The traces are normalized, considering the intensity differences of the two fluorescence channels, and are molar proportionate. Dashed lines indicate the peak retention of Slo-TS determined in separate experiments. The black trace (right chromatogram) indicates a Gaussian fit to the primary Slo1–β3 species, separating them from lower-order (disassembled) protein complexes. Arrows from last row of A highlight origin of material for each chromatogram in B. **(C)** Molar ratio of β2RF-β3Gρ in cellular lysates, total Slo1 isolates, and in the triply purified ternary complex in the two transfection ratios tested (TR1-1:1:1 and TR2-1:0.2:1.8; as described in the main text). **(D)** Fraction of total Slo1–β complexes that are binary (bin: single type of β subunit, β2 or β3) or ternary (ter: β2 and β3) in the two cases of transfection. In C and D, error bars indicate SD of 3 independent experiments.

during the initial paper describing structures of α+β4 BK complexes (Tao and MacKinnon, 2019), the authors noted that they did not see any evidence for the presence of less than four β4 subunits in their complexes, suggesting that rules for assembly may differ among different β subunits. The approaches utilized in the present work should prove useful in addressing this question for other β subunit combinations for BK channels.

To confirm that both β2 and β3a subunits can be present in the same BK channel complex, single-channel recordings directly demonstrated that, with coexpression of β2 and β3a cRNA, recovery from inactivation of single channels exhibited features of β2-mediated inactivation in some sweeps while, in other sweeps, β3a-mediated inactivation. Although the specific stoichiometry of β2:β3a in the single channels was not determined, the distribution of likelihoods of observing β3a-type tails openings during recovery from inactivation was consistent with patches containing either 1:3, 2:2: 1:3, or 0:4 β2:β3a combinations. Furthermore, biochemical assessments of BK channel composition using differentially fluorescently tagged β2 and β3a subunits expressed in HEK cells directly revealed that ternary complexes of α:β2:β3 form, that that ternary complex comprises a substantial population of all complexes, and that relative amounts of ternary complexes as well as mean stoichiometry of the ternary complexes can be tuned by expression levels of the subunits.

### Evaluation of models of assembly
The use of binomial distributions has been a powerful tool in the evaluation of models of protein subunit assembly (Fang et al., 2014) and has been applied to evaluation of a variety of ion channel complexes. For subunits of inwardly rectifying K⁺ channels (Glowatzki et al., 1995; Fakler et al., 1996) and glutamate-activated AMPA receptors (Mansour et al., 2001; Yu et al., 2021), evidence suggests that different isoforms of the pore-forming subunits assemble non-stochastically to form a functional channel. Moreover, in the pLGIC family of channels, it is well known that specific receptor subtypes adopt a preferential configuration, which is an important instance of non-stochasticity. However, in context of the channel-auxiliary subunit-dependent assemblies, non-stochastic assembly does not appear to be common. While cryo-EM reconstructions of native glutamate receptors identify receptors with distinct auxiliary subunits (TARP and CNIH) assembled into a single complex (Yu et al., 2021), the structures are able to reliably define the composition of perhaps only 5–10% of the pool of purified protein (corresponding to the number of particles that can be reliably aligned and refined to high resolution), which in itself might only represent a fraction of the total protein present in the cell. That specific assemblies exist in cells, where the expression of the individual subunits is genetically tuned, does not violate the prediction of binomial or stochastic assembly. Type 1 and 2A sulfonylurea receptors randomly assemble into KATP channel complexes to form heteromeric complexes in accordance with binomial statistics (Chan et al., 2008).

Despite the value of the binomial distribution for many biological questions, there are likely many situations in which other tools are required. Biochemical approaches increasingly have revealed examples of channel complexes with more than two components, e.g., ternary complexes of Kv4.2, with Kchip3 and DPP10 (Jerng et al., 2005; Jerng and Pfaffinger, 2012) and assembly of different KCNE subunits in complex with KCNMQ1 (Morin and Kobertz, 2007). However, as yet tools to address whether assembly is random or constrained do not seem to have been applied. To our knowledge, trinomial distribution analysis has not been used to evaluate questions pertinent to the assembly of molecules. Yet, it has been usefully applied to a variety of topics ranging from estimations of allele frequencies (Lynch, 2009) as an extension of the Hardy–Weinberg principle (Hardy, 1908; Edwards, 2008), financial engineering, stochastic modeling, and modeling outcomes of particle interactions or the particle decay processes in particle physics. In the present case, that a trinomial analysis could be tested in regard to β2 and β3a subunit assembly depended critically upon the unique properties conveyed on BK channels by the β2 and β3a subunits. However, the general approach could be potentially extensible to other cases if one can engineer functional reporters into the components being evaluated.

### Possible underpinnings of higher likelihood of β2 presence in ternary complexes
That equimolar expression of β2:β3a cRNA in oocytes or transfection with equimolar β2:β3a DNA in HEK cells results in a higher mole fraction of β2 subunits in the resulting ternary complexes was unexpected. However, even at the earliest biochemical measurements of molar ratios of β2 and β3a subunits in cell lysates, there is an excess of β2 subunits relative to β3a on the order of three to sixfold. Since the final ratios of β2:β3a in complexes tend to track roughly with the initial ratio in bulk lysates, for the present we prefer the view that something rate limiting in the translation process limits synthesis of β3a. In this regard, it is perhaps significant that when HEK cells are transfected with a ratio of 0.2:1.8 β2:β3a, with β3a message expected to be in ninefold excess, the relative abundance of β3a subunits in the final ternary complexes does not exhibit a similar excess, suggesting that too much β3a transcript may have a negative impact on the generation of the final protein product. Obviously, this work does not address whether there may be similar processes at play in native cells. However, that similar constraints on the eventual likelihood of β2 vs. β3a presence in a BK channel complex occur in both oocytes and HEK cells suggests it may represent differences in something intrinsic to the subunit sequences.

### Looking for BK channels of mixed β2 subunit composition in native cells
The present results provide clear proof-in-principle that different β subunit isoforms can coassemble in the same BK channels if they are both expressed in a cell. For an apparently well-studied channel, it is remarkable how poorly the subunit composition of BK channels in many tissues is understood. Despite great interest in the diversity of BK channel function that is generated by the β, γ, and LINGO families of regulatory subunits (Torres et al., 2007; Gonzalez-Perez and Lingle, 2019; Dudem et al., 2020) and quite detailed biophysical and mechanistic evaluation of the regulatory effects (Cox and Aldrich, 2000;

Contreras et al., 2012; Castillo et al., 2015; Gonzalez-Perez et al., 2018), knowledge of where such subunits are found has not advanced to quite the same extent. There are multiple complicating factors that have hindered progress. For example, for β subunits, there is evidence that specific motifs may influence subunit trafficking (Toro et al., 2006), even to intracellular organelles (Zarei et al., 2007), such that the presence of a message and protein for a subunit in a specific tissue may not reflect cell surface expression. Furthermore, BK regulatory subunits are expressed at low message abundance, even in cases where functional hallmarks of subunit presence are unambiguous.

In some cases, unique functional properties conferred by regulatory subunits have facilitated identification of where they may be expressed. For example, for γ1-containing BK channels, the over –100 mV gating shifts produced by γ1 (Yan and Aldrich, 2010, 2012; Gonzalez-Perez et al., 2014) have enabled confirmation of cellular expression in a number of types of secretory epithelial cells (Yang et al., 2017; Gonzalez-Perez et al., 2021). The more modest gating shifts produced by β1 subunits match with the more negative gating range of BK channels in smooth muscle, while the inactivation properties of β2-containing BK channels (Wallner et al., 1999; Xia et al., 1999) match well with native BK channel inactivation in chromaffin cells (Solaro et al., 1995; Solaro et al., 1997; Martinez-Espinosa et al., 2014), in some pancreatic β cell–derived cell lines (Li et al., 1999), and also some DRG neurons (Li et al., 2007). Similarly, some features of β4-containing BK channels (Brenner et al., 2000), including resistance to block by iberiotoxin, has been associated with neuronal BK channels (Brenner et al., 2005). However, such studies largely reflect the very tip of the iceberg of where BK β subunits are expressed, whether the functional properties have been adequately defined to support identification of potential regulatory subunits, let alone address whether there is overlap in expression.

The challenges of using message as an indicator are apparent in work on mouse adrenal glands and adrenal medullary chromaffin cells. Quantitative PCR reported the presence of Kcnmb2 message in whole adrenal glands (Martinez-Espinosa et al., 2014), although at levels less than Kcnmb4 expression and two orders of magnitude less than the Kcnma1 message. Expression of a message of a regulatory subunit at such small fractions of the associated pore-forming subunit might raise questions about its relevance, but it is unambiguous that inactivation of BK channels in mouse chromaffin cells arises from expression of β2 subunits (Martinez-Espinosa et al., 2014).

Relevant to the present work, knowledge about loci of expression of Kcnmb3 subunits is exceedingly limited, with no reports of BK currents in native cells that exhibit any of the characteristic features of β3-containing BK channels. At this point, all we can say is that, should β2 and β3a subunits be expressed in the same cell, there should be ternary complexes. Ultimately, the net result of the formation of ternary complexes would a new nuance to observed functional diversity of BK channels available to modulate cellular excitability.

### Data availability
All data underlying main or supplementary figures are available from the corresponding authors upon reasonable request.

## Acknowledgments
Olaf S. Andersen served as editor.

This work was supported by the National Institutes of Health (NIH) GM-118114 to Christopher J. Lingle and NIH GM-145719 and start-up funds from the Department of Molecular Physiology and Biophysics, The University of Iowa, to Sandipan Chowdhury.

Author contributions: Yu Zhou: conceptualization, data curation, formal analysis, investigation, methodology, software, validation, visualization, and writing—original draft, review, and editing. Vivian Gonzalez-Perez: investigation. Xiao-Ming Xia: conceptualization, data curation, investigation, and resources. Gopal S. Kallure: formal analysis, investigation, validation, and visualization. Sandipan Chowdhury: funding acquisition, investigation, methodology, project administration, resources, supervision, validation, visualization, and writing—original draft, review, and editing. Christopher J. Lingle: conceptualization, data curation, formal analysis, funding acquisition, methodology, project administration, resources, supervision, validation, visualization, and writing—original draft, review, and editing.

Disclosures: The authors declare no competing interests exist.

Submitted: 15 September 2025

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

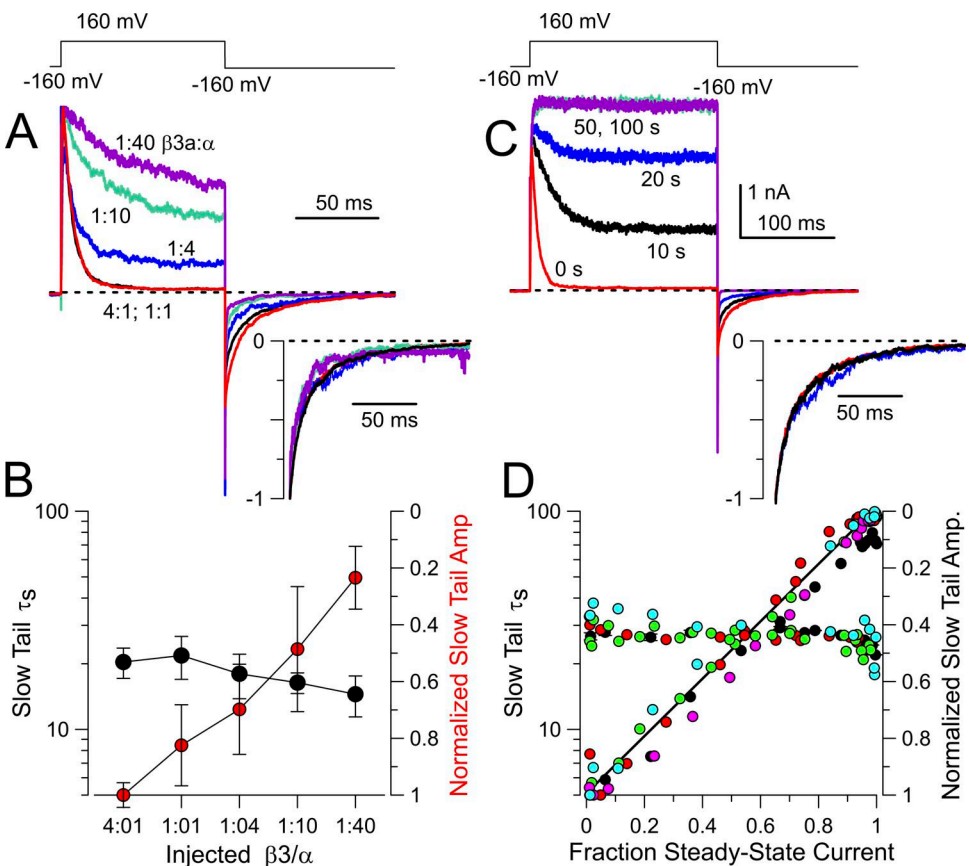

Figure S1. **β3a-mediated slow tail time constants do not change as number of β3a subunits in a BK channel complex is reduced. (A)** Currents are from 4 different patches each obtained from the indicated nominal injection ratios of β3a:α message. For each patch, trypsin was later applied to remove inactivation, and the slow tail amplitude was normalized to the outward BK current at +160 mV. The increase in non-inactivating current as the mole fraction of β3a is reduced reflects the decrease in the average number of β3a subunits per BK channel. The inset shows normalized slow tails for this set of patches, indicative that the slow tail current is not appreciably changing. **(B)** Mean (+SD) time constants for the given injection ratios reveal no major change with β3a stoichiometry, although the normalized slow tail amplitude (red symbols) is markedly reduced. **(C)** Example traces are from a single β3a+α patch exposed to trypsin for the indicated cumulative durations. The inset shows normalized slow tail currents at 0, 10, and 20 s of trypsin digestion. **(D)** Individual slow tail time constants from 5 patches at different times of trypsin digestion are plotted as a function of measured $f_{ss}$, for the same set of patches. Each individual patch is plotted with a given color for both the time constant and the amplitude.

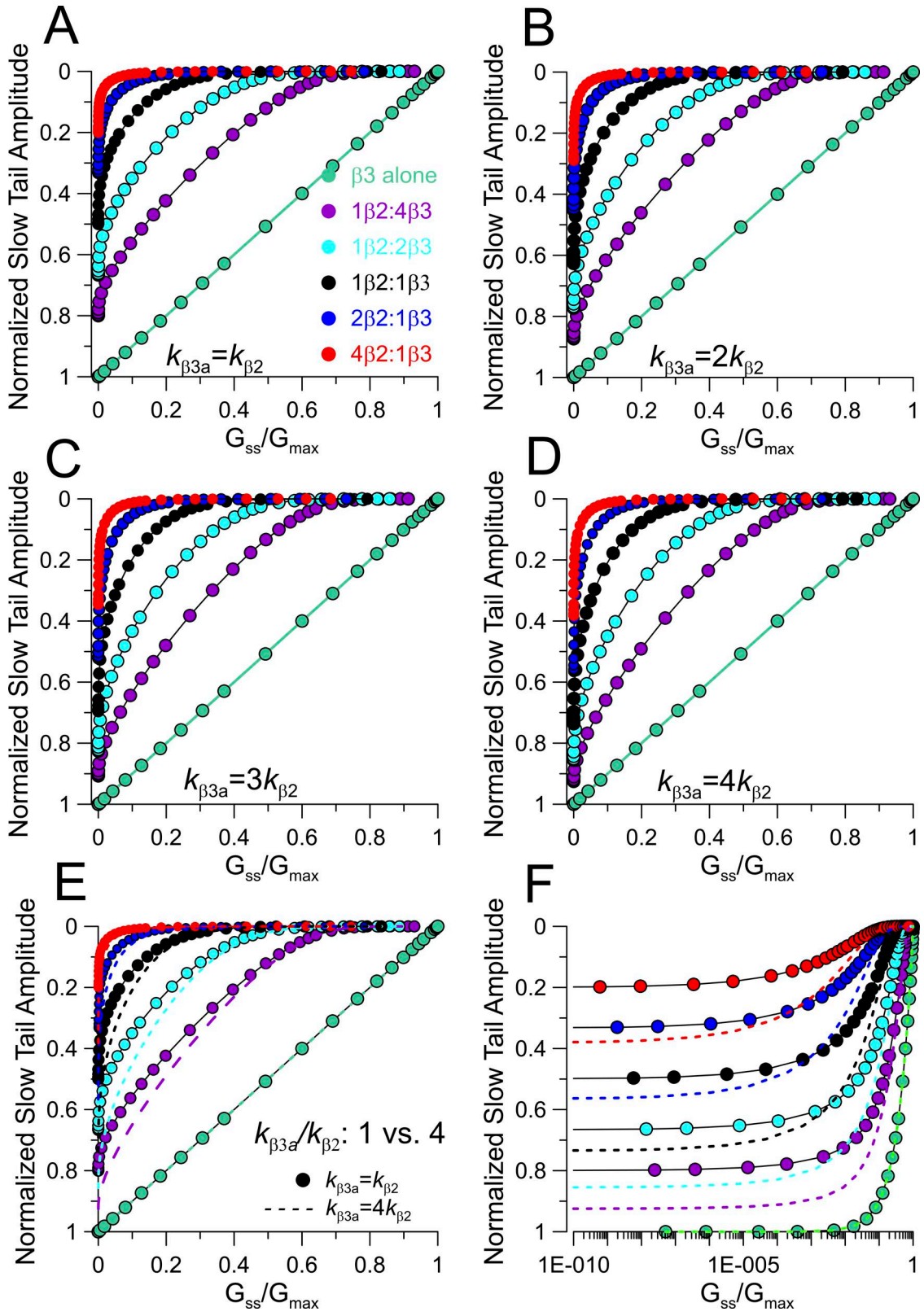

Figure S2. **Impact of differential β3a vs. β2 inactivation likelihoods ($L_{β3a}$, $L_{β2}$) on relationship between slow tail amplitudes and $f_{ss}$ for the random mixing model. (A)** Plot of the $A_{ST}$ vs. $f_{ss}$ relationships for the indicated β2:β3a ratios for the case of equal inactivation rates ($k_{β3a} = k_{β2}$). **(B)** $A_{ST}$ vs. $f_{ss}$ with $k_{β3a} = 2k_{β2}$. **(C)** $A_{ST}$ vs. $f_{ss}$ with $k_{β3a} = 3k_{β2}$. **(D)** $A_{ST}$ vs. $f_{ss}$ with $k_{β3a} = 4k_{β2}$. **(E)** Overlaid curves for $k_{β3a}/k_{β2}$ of either 1 or 4 (dotted lines). **(F)** Same as E on log axis. Overall, although changes in relative likelihood of β3a vs. β2 inactivation alter the relationship between normalized slow tail amplitude ($A_{ST}$) and steady-state current amplitude ($G_{ss}/G_{max}$) during trypsin digestion at different initial β2:β3a ratios, the overall impact is rather modest and does not alter the basic curvature in the $A_{ST}$ vs $G_{ss}$ relationship predicted by the random mixing model.

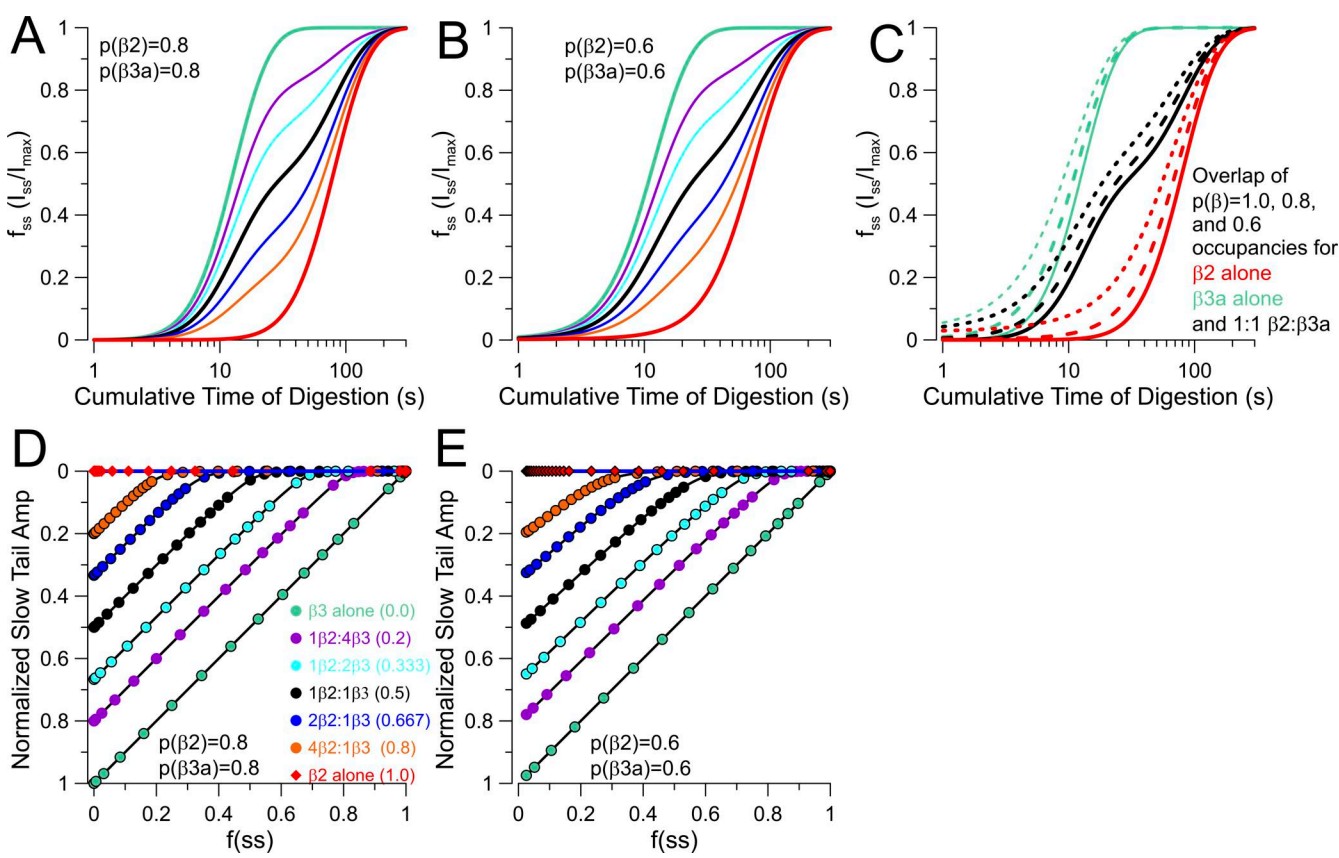

Figure S3. **Inclusion of partial occupancies in simulations of segregated assembly models does not mimic predictions of random mixing. (A)** Digestion time course for segregated β2 and β3 a populations, each with partial occupancy of P = 0.8, for β3 a/β2 ratios of 1, 0.8, 0.667, 0.5, 0.333, 0.2, and 1.0. **(B)** As in A, but with P = 0.6. **(C)** Comparison of digestion time course for β3a/(β3a+β2) ratios of 1(green), 0.5 (black), and 0 with occupancies within each segregated population of 1.0, 0.8, and 0.6. **(D)** Plot of the $A_{ST}$ vs. $f_{ss}$ relationships for β3a/(β2+β3a) ratios of 1.0 (green), 0.8, 0.667, 0.5 (black), 0.333, 0.2, and 0 (red) for the case of partial occupancies of 0.8. **(E)** As in D, but with partial occupancies of 0.6. Partial occupancies do not alter the absolutely linear dependence of slow tail reduction on fractional removal of β3a N termini.

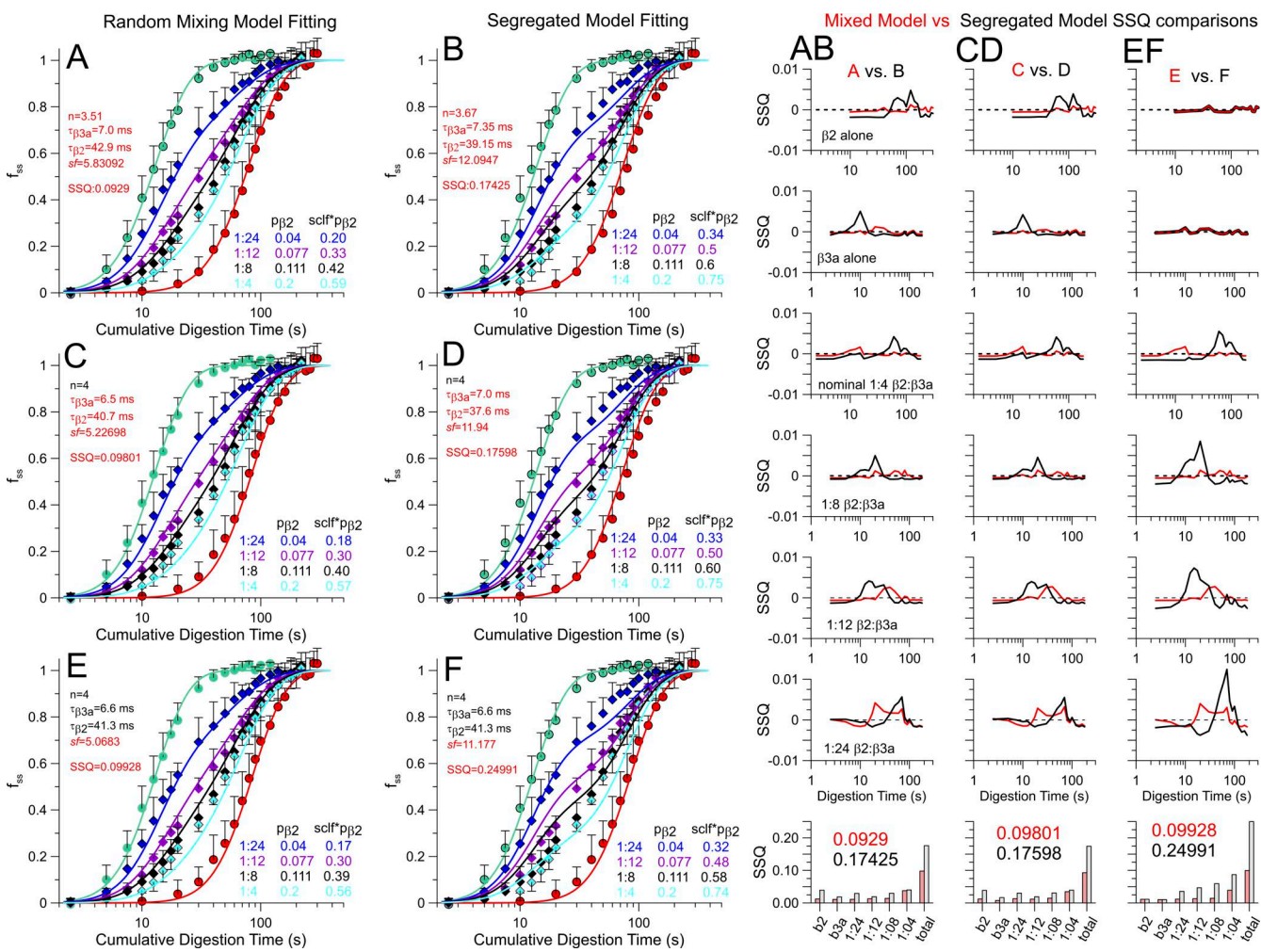

Figure S4. **Comparison of fits to digestion time course with different fitting assumptions. (A and B)** Best fit of random mixing model (left) and segregated model (right) with all parameters ($\tau_{\beta3a}$, $\tau_{\beta2}$, n, and sf) unconstrained. Table on the lower right of each panel displays the nominal and effective mole fraction of β2. (AB) The column shows, for each time point in A and B, the SSQ for the random mixing model (in red) and the segregated model (in black), with on bottom the summer SSQ for each digestion time course and then the total. **(C and D)** Best fit for each model with the additional constraint that n, the parameter for average number of IDs (n) per channel in the population, was set to 4. (CD) Comparison of SSQ at different time points across all conditions of injection ratios. **(E and F)** As in the main text, best fits for each model with only sf being a free parameter. (EF) SSQ comparisons at each time point and for each injection ratio. We note that, for the random mixing model, the change in SSQ going from 4 free parameters (A:0.0929) to 1 free parameter (E:0.0993) is minor compared with the difference between the random mixing model with 1 free parameter (E:0.0993) and the segregated model with 4 free parameters (B:0.1743). Irrespective of the fitting constraints, the random mixing model provides a better fit than the segregated model with identical numbers of free parameters. SSQ, sum of squares.

