## [Peer Review File · The Journal of General Physiology]

beta2 and beta3a regulatory subunits can coassemble in the same BK channels

Yu Zhou, Vivian Gonzalez-Perez, Xiao-Ming Xia, Gopal Kallure, Sandipan Chowdhury, and Christopher Lingle

Corresponding Author(s): Christopher Lingle, Washington University in St. Louis School of Medicine and Yu Zhou, Washington Univ. School of Medicine

Review Timeline:

Submission Date:	September 15, 2025
Editorial Decision:	October 9, 2025
Revision Received:	October 21, 2025
Editorial Decision:	October 22, 2025
Revision Received:	October 23, 2025

Editor: Olaf Andersen

Transaction Report:

DOI: <https://doi.org/10.1085/jgp.202513890>

October 9, 2025

Dr. Christopher J Lingle
Washington University in St. Louis School of Medicine
Anesthesiology
Box 8054
St. Louis, MO 63110

Re: 202513890

Dear Chris,

Thank you for submitting your manuscript, entitled "beta2 and beta3a regulatory subunits can coassemble in the same BK channels" to JGP. Your manuscript has now been seen by 3 reviewers, whose comments are appended below. You will see that the reviewers were very enthusiastic about the study and its potential impact and raised only minor concerns/suggestions that should nevertheless be addressed prior to further consideration of the manuscript at JGP. Based on my own reading, I agree with Reviewer 1 that you have earned the right to make some larger statements, where it would be helpful if you include the examples listed by Reviewer 3. With respect to Reviewer 3's comment on Figure 9, could you use the same color scheme (for the beta subunits) as in Figure 4.

We hope that you will be able to submit a revised manuscript that addresses these points, which we believe will pose no problems, and which may be re-reviewed. In addition, please do not hesitate to contact me (via the editorial office) if you feel that a discussion of the reviewers' and editors' comments would be helpful.

Please submit your revised manuscript via the link below, along with a point-by-point letter that details your response to the reviewers' and editor's summary, as well as a copy of the text with alterations highlighted (boldfaced or underlined). If the article is eventually accepted, it would include a 'revised date' as well as submitted and accepted dates. If we do not receive the revised manuscript within one year, we will regard the article as having been withdrawn. We would be willing to receive a revision of the manuscript at a later time, but the manuscript will then be treated as a new submission, with a new manuscript number.

Please pay particular attention to recent changes to our instructions to authors in the following sections: Data presentation, Blinding and randomization and Statistical analysis, under Materials and Methods, as shown here: <https://rupress.org/jgp/pages/submission-guidelines#prepare>. Re-review will be contingent on inclusion of the required information (including for data added during revision) and demonstration of the experimental reproducibility of the results. Also, To improve the reproducibility of published content, we have partnered with SciScore. Authors are prompted in eJP to copy and paste the Materials and Methods section of their manuscript for a SciScore assessment when submitting their revised manuscript. Authors are encouraged (not required) to further revise their Materials and Methods if the SciScore is below 4. More information can be found here: <https://rupress.org/jgp/pages/submission-guidelines#sciscore>.

Please note, JGP now requires authors to submit Source Data used to generate figures containing gels and Western blots with all revised manuscripts (when applicable). This Source Data consists of fully uncropped and unprocessed images for each gel/blot displayed in the main and supplemental figures. If your paper includes cropped gel and/or blot images, please be sure to provide one Source Data file for each figure that contains gels and/or blots along with your revised manuscript files. File names for Source Data figures should be alphanumeric without any spaces or special characters (i.e., SourceDataF#, where F# refers to the associated main figure number or SourceDataFS# for those associated with Supplementary figures). The lanes of the gels/blots should be labeled as they are in the associated figure, the place where cropping was applied should be marked (with a box), and molecular weight/size standards should be labeled wherever possible. Source Data files will be made available to reviewers during evaluation of revised manuscripts and, if your paper is eventually published in JGP, the files will be directly linked to specific figures in the published article.

Source Data Figures should be provided as individual PDF files (one file per figure). Authors should endeavor to retain a minimum resolution of 300 dpi or pixels per inch. Please review our instructions for export from Photoshop, Illustrator, and PowerPoint here: <https://rupress.org/jgp/pages/submission-guidelines#revised>

Whilst you are revising your manuscript, we ask that you consider whether you have any artwork that might be suitable for the cover of JGP. Microscopy images are particularly good for cover artwork, but other types of image can be very effective, so we encourage you to be creative. Please don't restrict yourself to images from the paper; an image that is relevant to the work described would be just as suitable. Images should be a minimum resolution of 300 dpi. To see recent examples, visit the following page and click on 'Show covers? Yes': <https://jgp.rupress.org/content/by/year>

Thank you for submitting your interesting research to JGP.

Please submit your revised manuscript, and any associated files, via this link:
Link Not Available

Sincerely,

Olaf
On behalf of Journal of General Physiology

Journal of General Physiology's mission is to publish mechanistic and quantitative molecular and cellular physiology of the highest quality; to provide a best-in-class author experience; and to nurture future generations of independent researchers.

Reviewer #1 (Comments to the Authors):

This tour-de-force study by Zhou shows asks the important question of whether different beta subunits for BK channels can co-assemble in single channel complexes. They answer the question for beta2 and beta3a modulatory subunits. Yes, these two beta subunit types can co-assemble in individual BK channels. The results appear solid, the findings highly important, and the paper clearly written, mechanistic, and well suited to JGP.

I have no major concerns, but only a few minor comments to potentially tweak the writing.

Abstract - last sentence. Indicate the three types of subunits that form ternary complexes, as you do on pg. 5 line 5.

Optional: Pg. 7 "and k and T have the usual physical meanings." Is this in contrast to unusual physical meanings? Why not just state $kT = ??$ eomV and indicate the temperature. Google AI thinks kT (at 37 deg. C) is 26.6 eomV. You could state kT for your experimental temperature. Why does it matter. Because knowing the values allows one an immediate estimate of the voltage sensitivity in mV per efold change in P_o at low levels of P_o .

Pg. 12 ln 10: used previously (add a reference here)

Pg 34, line 12 typo "bene"

Figure S2. Indicate to the reader in a line or two here what the impact (finding) of this figure is.

Figure S4. Part A. Suggest for consistency that the A figure be labelled Random Mixing Model Fitting. Add a conclusion sentence at end of the figure legend.

Optional. This study shows that random mixing of various types of beta subunits can occur in single channel complexes and that the examined beta subunits can retain their individual inactivation behavior (pg. 28) independent of the presence of the other isoform. Never missing a chance for some general observations or exceptions to bold conclusions, a few lines as to what extent the behavior of subunits is independent (or not) of other subunits as a general (nothing is of course general) rule might be worth making. Geng et al. 2023 JGP reported apparent random assembly of a WT alpha BK and mutant alpha BK and independent action of WT and mutant alpha subunits (for the examined mutants) on the voltage sensitivity and conductance. There are likely many more examples in the literature (for example beta and gamma BK subunits as exceptions?).

The conclusions are well supported for the two types of beta subunits explored. In real life would human alpha, human beta2, and human beta3a form ternary complexes? The authors may wish to add a sentence indicating their best guess on this question. Maybe it is already in the manuscript.

Supplemental material is justified.

Reviewer #2 (Comments to the Authors):

This is a very well-written and clear paper, although complex. There are very few, if any other laboratories that could complete this sort of study, especially regarding electrophysiology. The combination of electrophysiology and biochemical approaches to identify that a BK channel complex can have both beta2 and beta3 subunits is powerful. The authors have embedded throughout the manuscripts appropriate caveats. Although I think the manuscript is sufficient for publication without any further experimental work, one potential additional experiment would be to demonstrate that other combinations of beta subunits could also co-assemble using the biochemical approach (although perhaps not physiologically relevant). If these data are available, I would recommend including them; otherwise, what is present in this tour-de-force manuscript is already more than sufficient.

Reviewer #3 (Comments to the Authors):

This is a wonderful paper that was a joy to read. The techniques employed are masterful and the logical flow is straightforward. Despite the complexity, I found the paper easy to follow and could easily extract detail.

What follows are minor comments that could improve a wonderful manuscript further. Congratulations, this is the best work I reviewed for a long time.

1)

it's a shame to have Figure 9 looking so different from the rest. There are some errors in the conversion, but I would also use the same font and try to reach the clarity of the other figures.

2)

I was looking for information about the probabilities used in each model but didn't find it easily. The biochemistry suggests that the assembly has a very minor bias. Is it worth using specific probabilities? Perhaps the authors could comment on this in the manuscript.

3)

p8 You mention other cases where the binomial model has been used, but I found the examples given are curious. One example given is the Ding paper (on BK) and the other is a paper from Mackinnon on Shaker. Of course, this model was used for subunit assembly in other interesting cases outside of the narrow potassium channel world. Perhaps the authors could put their work in a bit wider overall context:

For "pore forming subunits":

Glutamate Receptors

Mansour, M, Nagarajan, N, Nehring, RB, Clements, JD, and Rosenmund, C., (2001) Heteromeric AMPA receptors assemble with a preferred subunit stoichiometry and spatial arrangement.

TRP channels

<https://www.nature.com/articles/ncomms2257>

For auxiliary proteins:

Sulfonylurea Receptors Type 1 and 2A Randomly Assemble to Form Heteromeric KATP Channels of Mixed Subunit Composition

<https://doi.org/10.1085/jgp.200709894>

(in JGP!!!)

Glutamate receptors (and maybe others) show non-random assembly with auxiliary subunits:

Yu, J, Rao, P, Clark, S, Mitra, J, Ha, T, and Gouaux, E., (2021) Hippocampal AMPA receptor assemblies and mechanism of allosteric inhibition. DOI: 10.1038/s41586-021-03540-0

"The native hpAMPA complexes do not have the bias of artificial, engineered complexes and show how the functional properties of AMPARs are sculpted by the non-stochastic assembly of receptor and auxiliary protein components."

Thank you for submitting your manuscript, entitled "beta2 and beta3a regulatory subunits can coassemble in the same BK channels" to JGP. Your manuscript has now been seen by 3 reviewers, whose comments are appended below. You will see that the reviewers were very enthusiastic about the study and its potential impact and raised only minor concerns/suggestions that should nevertheless be addressed prior to further consideration of the manuscript at JGP. Based on my own reading, I agree with Reviewer 1 that you have earned the right to make some larger statements, where it would be helpful if you include the examples listed by Reviewer 3. With respect to Reviewer 3's comment on Figure 9, could you use the same color scheme (for the beta subunits) as in Figure 4.

Reviewer #1 (Comments to the Authors):

This tour-de-force study by Zhou shows asks the important question of whether different beta subunits for BK channels can co-assemble in single channel complexes. They answer the question for beta2 and beta3a modulatory subunits. Yes, these two beta subunit types can co-assemble in individual BK channels. The results appear solid, the findings highly important, and the paper clearly written, mechanistic, and well suited to JGP. I have no major concerns, but only a few minor comments to potentially tweak the writing.

Very much appreciated.

Abstract - last sentence. Indicate the three types of subunits that form ternary complexes, as you do on pg. 5 line 5. **DONE**

Optional: Pg. 7 "and k and T have the usual physical meanings." Is this in contrast to unusual physical meanings? Why not just state $kT = ?? \text{ eomV}$ and indicate the temperature. Google AI thinks kT (at 37 deg. C) is 26.6 eomV. You could state kT for your experimental temperature. Why does it matter. Because knowing the values allows one an immediate estimate of the voltage sensitivity in mV per efold change in P_o at low levels of P_o . The following is now inserted: "with kT set to 25.7 meV at 25C".

Pg. 12 ln 10: used previously (add a reference here)

Pg 34, line 12 typo "bene" **FIXED**

Figure S2. Indicate to the reader in a line or two here what the impact (finding) of this figure is. **DONE**

Figure S4. Part A. Suggest for consistency that the A figure be labelled Random Mixing Model Fitting. Add a conclusion sentence at end of the figure legend. **DONE**

Optional. This study shows that random mixing of various types of beta subunits can occur in single channel complexes and that the examined beta subunits can retain their individual inactivation behavior (pg. 28) independent of the presence of the other isoform. Never missing a chance for some general observations or exceptions to bold conclusions, a few lines as to what extent the behavior of subunits is independent (or not) of other subunits as a general (nothing is of course general) rule might be worth making. Geng et al. 2023 JGP reported apparent random assembly of a WT alpha BK and mutant alpha BK and independent action of WT and mutant alpha subunits (for the examined mutants) on the voltage sensitivity and conductance. There are likely many more examples in the literature (for example beta and gamma BK subunits as exceptions?).

We have now included Geng et al as another example in which binomial considerations resulted in the conclusion of random assembly of a WT and mutant subunit. This was an unfortunate omission. Regarding whether the independent behavior of $\beta 2$ and $\beta 3a$ is the rule for other BK β s, we have intentionally skirted around this issue, although we do point out some aspects of $\beta 4$ that suggest it may behave differently.

From additional experiments, we suspect the situation may be quite different with some of the other BK β subunits. We have made attempts to use our trinomial assembly approach to test other combinations, but it will be quite some time before such experiments are completed. Furthermore, the Chowdhury lab has data on combinations assessed biochemically. The short answer is that independent assembly is NOT a general rule, and partial stoichiometries (as observed in Wang et al., 2002) may not necessarily be the rule for all betas. These other observations would add an important layer of complexity to the overall topic. In terms of the present paper, we think any brief mention of what we know about other combinations would not be helpful and only seem like a poorly documented teaser. We

definitely do not wish to imply that independent assembly is the rule, but in this case and with the analytic tools we employed, random independent assembly provides the best explanation. We have indicated in the discussion that this may not be generalizable. Although the reviewers are correct that one would like to know about other combinations, a lot more work will be necessary to sort this out and we feel it would be premature to say anything more at this point. In essence, the approaches in the present manuscript outline the strategies that can be used to address these issues in the future (hopefully soon).

The conclusions are well supported for the two types of beta subunits explored. In real life would human alpha, human beta2, and human beta3a form ternary complexes? The authors may wish to add a sentence indicating their best guess on this question. Maybe it is already in the manuscript.

We decline to say more on this issue in the manuscript. The challenge is that β 3a message is at very low abundance in almost all loci, except for olfactory bulb and also vomeronasal organ epithelium (unpublished results). We (VGP) have recorded patches from VNO sensory epithelial cells in which BK channels show recovery from inactivation with either β 2 or β 3a type behavior. However, because there are multiple channels in the patches, at present we cannot definitively say that both subunits are in the same channel. We think they are, but this is a topic for future work.

Supplemental material is justified.

Reviewer #2 (Comments to the Authors):

This is a very well-written and clear paper, although complex. There are very few, if any other laboratories that could complete this sort of study, especially regarding electrophysiology. The combination of electrophysiology and biochemical approaches to identify that a BK channel complex can have both beta2 and beta3 subunits is powerful. The authors have embedded throughout the manuscripts appropriate caveats. Although I think the manuscript is sufficient for publication without any further experimental work, one potential additional experiment would be to demonstrate that other combinations of beta subunits could also co-assemble using the biochemical approach (although perhaps not physiologically relevant). If these data are available, I would recommend including them; otherwise, what is present in this tour-de-force manuscript is already more than sufficient.

Again, thanks for the favorable comments. Regarding other combinations of beta subunits, please see the responses to Rev. #1 above.

Reviewer #3 (Comments to the Authors):

This is a wonderful paper that was a joy to read. The techniques employed are masterful and the logical flow is straightforward. Despite the complexity, I found the paper easy to follow and could easily extract detail.

What follows are minor comments that could improve a wonderful manuscript further. Congratulations, this is the best work I reviewed for a long time.

Thanks for the kind comments.

1) It's a shame to have Figure 9 looking so different from the rest. There are some errors in the conversion, but I would also use the same font and try to reach the clarity of the other figures.

We hope this was just an oversight in the file we uploaded. However, as an attempt to improve visualization, we have rearranged the Fig making it more vertical, which we think enhances the chromatograms and panels B-D which contain the actual data of interest.

2) I was looking for information about the probabilities used in each model but didn't find it easily. The biochemistry suggests that the assembly has a very minor bias. Is it worth using specific probabilities? Perhaps the authors could comment on this in the manuscript. We are uncertain to what probabilities the reviewer is referring. Implicit in the trinomial equation and the sum of two binomial distributions are terms for the relative mole fractions

of β_2 and β_{3a} that are defined by the injected ratios, although corrected by the “self” term that accounts for whatever it is that is impacting on the apparently intrinsic difference in relative efficacy of assembly (or expression) of each type of subunit in a complex. Thus, at an injection ratio of $\beta_2:\beta_{3a}$ of, say, 1:1, the nominal $p_{\beta_2}=0.5$ and $p_{\beta_{3a}}=0.5$. However, as I hope we had made clear, for a 1:1 injection ratio, the results indicate that β_2 has an approximately 6-fold higher likelihood of being assembled. The fact that, when we fit the digestion time courses for all injection ratios simultaneously, the self=6 term applies over all injection ratios provides some confidence that this is really some intrinsic difference in expression/assembly efficiency between the two subunits. Furthermore, the trend of this difference in efficiency was also qualitatively seen in the biochemical work, as we noted in the manuscript.

I want to mention that we have some preliminary control experiments in which we have paired coexpression of a β_2 subunit, with a β_2 subunit with a β_{3a} N-terminus. In that case, the cores of both subunits are β_2 and in that case the self term comes out very close to 1. Those results are not ready to be included in the present paper, but will be used with results on other subunit combinations. However, we mention that result here since it provides some confidence for our explanation that the self term is reflecting an intrinsic difference between subunits. We are not sure this answers the Reviewer’s concern about what probabilities are used, but in essence the probabilities are defined by the ratio of injected cRNA corrected by the self. In Figure S4, the nominal p_{β_2} vs corrected p_{β_2} are given in each panel from A to F, for each of the fit examples,

3) p8 You mention other cases where the binomial model has been used, but I found the examples given are curious. One example given is the Ding paper (on BK) and the other is a paper from Mackinnon on Shaker. Of course, this model was used for subunit assembly in other interesting cases outside of the narrow potassium channel world. Perhaps the authors could put their work in a bit wider overall context:

For "pore forming subunits":

Glutamate Receptors

Mansour, M, Nagarajan, N, Nehring, RB, Clements, JD, and Rosenmund, C., (2001) Heteromeric AMPA receptors assemble with a preferred subunit stoichiometry and spatial arrangement.

TRP channels

<https://www.nature.com/articles/ncomms2257>

For auxiliary proteins:

Sulfonylurea Receptors Type 1 and 2A Randomly Assemble to Form Heteromeric KATP Channels of Mixed Subunit Composition

<https://doi.org/10.1085/jgp.200709894>

(in JGP!!!) (Chan et al., 2008)

Glutamate receptors (and maybe others) show non-random assembly with auxiliary subunits:

Yu, J, Rao, P, Clark, S, Mitra, J, Ha, T, and Gouaux, E., (2021) Hippocampal AMPA receptor assemblies and mechanism of allosteric inhibition. DOI: 10.1038/s41586-021-03540-0

"The native hpAMPA complexes do not have the bias of artificial, engineered complexes and show how the functional properties of AMPARs are sculpted by the non-stochastic assembly of receptor and auxiliary protein components."

We thank the reviewer for correctly pointing out the numerous other situations in which a binomial model has been used to assess models of assembly. Although it is not feasible to point out all situations where such a model has been applied, we have now included some of those noted by the Reviewer and several others. In regards to the initial mention of the application of the binomial distributions pertinent to BK channels that is given in the Methods, we now focus there exclusively on specific cases pertinent to BK channels. However, embracing the Reviewer’s intent, we have added a brief new section to the Discussion that mentions many cases of application of binomial distributions to evaluate channel complexes, some supporting random assembly and some not, while then transitioning to a mention of the potential utility that trinomial distributions may have in selected cases of higher

order complexes. We thank the Reviewer and the Editor for this suggestion. Although we might certainly like to think that the trinomial approach will have applicability to other similar problems, we think one will have to be even more clever in coming up of ways of developing unique reporters that might allow the trinomial distribution to be utilized.

Dr. Christopher J Lingle
Washington University in St. Louis School of Medicine
Anesthesiology
Box 8054
St. Louis, MO 63110

Re: 202513890R1

Dear Chris,

I am pleased to let you know that your manuscript, titled "beta2 and beta3a regulatory subunits can coassemble in the same BK channels" is scientifically acceptable for publication in Journal of General Physiology. Formal acceptance will follow when it is modified in accordance with our editorial policies.

Please note items that need attention are listed at the bottom of this email (under 'manuscript formatting checklist'). Please also be sure to include a letter addressing the reviewers' comments point-by-point (if applicable) and a copy of the text with alterations highlighted (boldfaced or underlined). Your manuscript should be a double-spaced MS Word file and include editable tables, if appropriate.

Lastly, JGP requires a data availability statement for all research article submissions. These statements will be published in the article directly above the Acknowledgments. The statement should address all data underlying the research presented in the manuscript. Please visit the JGP instructions for authors for guidelines and examples of statements at <https://rupress.org/jgp/pages/editorial-policies#data-availability-statement>.

Please submit your final files via this link:
Link Not Available

Thank you for choosing to publish your research in JGP and please feel free to contact me with any questions.

Sincerely,

Olaf
On behalf of Journal of General Physiology

Journal of General Physiology's mission is to publish mechanistic and quantitative molecular and cellular physiology of the highest quality; to provide a best in class author experience; and to nurture future generations of independent researchers.

Manuscript formatting checklist:

- MS Word document of text needed (including editable tables)
- MS Word document of supplemental text needed, if applicable (including figure legends and editable tables)
- Brief Statement describing supplementary information needed, if applicable (in subsection at end of Materials & Methods)
- Please include a data availability statement preceding the Acknowledgments section. Please see <https://rupress.org/jgp/pages/editorial-policies#data-availability-statement>
- Figures created at sufficient resolution and in acceptable format (including supplemental if applicable. And no JPGs). If working in Illustrator, we prefer .ai or .eps file format. If working in Photoshop please use 600dpi/1000dpi .tiff or .psd file format. Minimum resolution at estimated print size: Minimum resolution for all figures is 600 dpi. For figures that contain both photographs and line art or text, 600 dpi is highly recommended. Figures containing only black and white elements (line art, no color, and no gray) should be 1,000 dpi. Maximum figure size is 7 in wide x 9 in high (17.5 x 22.8 cm) at the correct resolution. <https://jgp.rupress.org/fig-vid-guidelines>
- Supplemental figures, if any, conforming to same guidelines as manuscript figures (noted above)
- If images resemble one from a prior publications, the author must seek permissions (to reproduce or adapt) from the original publisher. [You can resubmit your paper while waiting to hear back from the original publisher but please keep us updated]
- All authors must complete a disclosure form prior to acceptance. A link to complete the form has been sent to all coauthors. Please provide the editorial office with updated email addresses if necessary